# Warsaw Breakage Syndrome associated DDX11 helicase resolves G-quadruplex structures to support sister chromatid cohesion

Janne J. M. van Schie[1,10], Atiq Faramarz[1,10], Jesper A. Balk[1], Grant S. Stewart [2], Erika Cantelli[3], Anneke B. Oostra[1], Martin A. Rooimans[1], Joanna L. Parish [2], Cynthia de Almeida Estéves[4], Katja Dumic[5], Ingeborg Barisic[6], Karin E. M. Diderich[7], Marjon A. van Slegtenhorst[7], Mohammad Mahtab [8], Francesca M. Pisani[8], Hein te Riele [3], Najim Ameziane[1,9], Rob M. F. Wolthuis [1✉] & Job de Lange [1✉]

Warsaw Breakage Syndrome (WABS) is a rare disorder related to cohesinopathies and Fanconi anemia, caused by bi-allelic mutations in *DDX11*. Here, we report multiple compound heterozygous WABS cases, each displaying destabilized DDX11 protein and residual DDX11 function at the cellular level. Patient-derived cell lines exhibit sensitivity to topoisomerase and PARP inhibitors, defective sister chromatid cohesion and reduced DNA replication fork speed. Deleting DDX11 in RPE1-TERT cells inhibits proliferation and survival in a TP53-dependent manner and causes chromosome breaks and cohesion defects, independent of the expressed pseudogene *DDX12p*. Importantly, G-quadruplex (G4) stabilizing compounds induce chromosome breaks and cohesion defects which are strongly aggravated by inactivation of DDX11 but not FANCJ. The DNA helicase domain of DDX11 is essential for sister chromatid cohesion and resistance to G4 stabilizers. We propose that DDX11 is a DNA helicase protecting against G4 induced double-stranded breaks and concomitant loss of cohesion, possibly at DNA replication forks.

[1] Section of Oncogenetics, Cancer Center Amsterdam and Department of Clinical Genetics, Amsterdam University Medical Centers, De Boelelaan 1118, 1081 HV Amsterdam, the Netherlands. [2] Institute of Cancer and Genomic Sciences, University of Birmingham, Edgbaston, Birmingham B15 2TT, UK. [3] Netherlands Cancer Institute, Division of Tumor Biology and Immunology, Amsterdam, The Netherlands. [4] Departamento de Genetica, Hospital Militar, Montevideo, Uruguay. [5] Department of Pediatric Endocrinology and Diabetes, University Hospital Centre Zagreb, University of Zagreb Medical School, Zagreb, Croatia. [6] Children's Hospital Zagreb, Center of Excellence for Reproductive and Regenerative Medicine, Medical School University of Zagreb, Zagreb, Croatia. [7] Department of Clinical Genetics, Erasmus Medical Center, Rotterdam, The Netherlands. [8] Istituto di Biochimica e Biologia Cellulare, Consiglio Nazionale delle Ricerche, Naples, Italy. [9] Present address: Centogene, Am Strande 7, 18055 Rostock, Germany. [10] These authors contributed equally: Janne J. M. van Schie, Atiq Faramarz. ✉email: r.wolthuis@amsterdamumc.nl; j.delange1@amsterdamumc.nl

Warsaw Breakage Syndrome (WABS) was discovered in 2010 in a patient displaying remarkable clinical overlap with Fanconi anemia (FA), a DNA damage syndrome characterized by impaired DNA crosslink repair, including growth retardation, microcephaly, and abnormal skin pigmentation, although bone marrow failure was not observed [1]. Both FA and WABS patient-derived cells exhibit mitomycin C (MMC)-induced chromosomal breaks but only WABS cells typically show spontaneous loss of sister chromatid cohesion at metaphase[2,3]. Such cohesion loss is significantly exacerbated by treatment with MMC or the topoisomerase I inhibitor camptothecin (CPT). Bi-allelic mutations in *DDX11* were identified as the WABS-underlying genetic defect. Previously, sixteen WABS individuals from twelve different families had been reported carrying bi-allelic DDX11 mutations[1,4–9].

DDX11 (also known as ChlR1) is one of the two human orthologues of yeast Chl1, a DNA helicase that was identified in a screen for mutants affecting chromosome segregation[10,11]. The other orthologue is DDX12p (also known as ChlR2)[12,13], a presumed pseudogene linked to a late-evolutionary duplication of a DDX11-containing region on chromosome 12. Parish and colleagues showed that DDX11 protein localizes to the nucleus and supports sister chromatid cohesion[14] which fits with later findings in WABS cells. DDX11 loss in mice is embryonically lethal[15,16], apparently contradicting the observed viability of yeast Chl1 mutants[10,11,17] and the relatively moderate symptoms of human WABS patients.

Sequence similarities classify DDX11 in a subgroup of DEAD/H box, ATP-dependent, super-family 2 (SF2) RNA or DNA helicases that contain an iron–sulfur cluster (Fe–S) between the Walker A and B boxes[18]. Other members include FANCJ, RTEL1, and XPD/ERCC2, all linked to genetic disorders characterized by defects in DNA repair mechanisms. Several in vitro studies[19–22] revealed that DDX11 unwinds duplex DNA in an ATP-dependent manner with 5′–3′ directionality, preferring DNA substrates with a 5′ single-stranded region. As recently reviewed[23], substrates include forked duplexes, 5′ flap structures (relevant in the processing of Okazaki fragments), three-stranded D-loops (early HR repair intermediates) and anti-parallel G-quadruplexes (G4s) that may form in stretches of G-rich DNA. Interestingly, mutating both *chl-1* and *dog-1* (the *C. elegans* orthologs of DDX11 and FANCJ) increased the number of deletions in poly-guanine tracts as compared to single dog-1 mutants, suggesting a potential role for DDX11 in resolving G4 or related poly-guanine duplex structures in vivo[24].

Yeast Chl1 mutant strains show increased sensitivity to DNA damaging agents including MMS and UV[17,25]. In addition, Chl1 supports the establishment of sister chromatid cohesion[26,27], a process coupled to DNA replication[28,29]. Unwinding of duplex DNA at the replication fork is performed by the Cdc45-MCM (mini-chromosome-maintenance)-GINS (go-ichi-ni-san) (CMG) complex. Specialized DNA polymerases synthesize DNA in conjunction with a homo-trimeric proliferating cell nuclear antigen (PCNA) sliding clamp that is loaded onto DNA by the replication factor C (RFC) complex. Other components of the DNA replication machinery include Tof1/Timeless, Csm3/Tipin, Mrc1/Claspin, and Ctf4/AND-1, which all play important roles in maintaining DNA replication fork stability by coordinating DNA unwinding with synthesis and the association with S phase checkpoint proteins (reviewed in ref. [30]).

*CHL1* interacts genetically with multiple components of the alternative RFC^CTF18 complex, including *CTF18*, *CTF8*, and *DCC1*, which are implicated in cohesion establishment, too[27,31–34]. Physical interactions were also reported between Chl1 and PCNA[35], Ctf4[36], and the 5′-flap endonuclease Fen-1[37]. Human DDX11 binds PCNA[20], WDHD1, POL δ[38], FEN1, promoting FEN1

activity, and RFC^CHTF18 which stimulates DDX11 helicase activity in vitro. Furthermore, a role for DDX11 was proposed in stabilization of the replication fork through an interaction with Timeless, a circadian rhythm regulatory gene. DDX11-Timeless complexes support DNA binding and helicase activity of DDX11 in vitro and promote sister chromatid cohesion[39–41]. Whether DDX11 helicase activity is critically involved in connecting DNA replication and sister chromatid cohesion, and, if so, which DNA substrates are targeted by DDX11 in vivo, remains a matter of debate. Hypothetically, DDX11 could facilitate sister chromatid entrapment into cohesin rings by resolving secondary DNA structures formed at replication sites, for instance in the lagging strand or in ssDNA formed during repair of stalled forks. Indeed, abolishing the ATPase activity in yeast Chl1[42] and in avian DDX11[43] resulted in loss of sister chromatid cohesion. However, dominant helicase-independent roles in cohesin loading have also been proposed[36,41].

Here, we identify seven new WABS patients from five different families. WABS-derived cells display loss of sister chromatid cohesion, increased sensitivity to CPT or PARP inhibitors and reduced replication fork speed. The investigated WABS cells retain residual DDX11 activity, originating from an unstable protein. Nevertheless, several of the identified mutant alleles are capable of rescuing sister chromatid cohesion when overexpressed in WABS patient cells, showing they are hypomorphic mutations with only partial loss of function. Inactivating DDX11 in diploid human epithelial cells causes TP53-dependent growth restriction and increased RAD51 dependence. While we show that DDX12p is expressed as mRNA, we observed no complementary role for DDX12p. DDX11 deficient cells display very high sensitivity to G4-stabilizing drugs, exceeding that of cells deficient for other putative G4 helicases FANCJ, BLM, and WRN. The DDX11 helicase domain is strictly required for sister chromatid cohesion. We propose therefore that DDX11 unwinds G4 structures that may arise when replication forks travel through G-rich regions, thus preventing breaks and promoting efficient DNA entrapment by cohesin rings. This likely contributes to genomic stability and cellular fitness.

## Results

**Clinical characterization of seven new WABS cases.** The first WABS patient was reported by our lab in 2010[1], which we now call WABS01. Here, we investigated cell lines derived from seven new WABS patients (WABS02–WABS08) with non-consanguineous parents and of different origins. WABS02 (male) is the second child of Dutch parents, initially diagnosed with Nijmegen Breakage Syndrome (NBS), although no *NBS1* mutations were found[44]. He showed growth retardation, microcephaly, deafness and abnormal skin pigmentation. WABS03 (male) is the second child of Uruguayan parents. He received pediatric intensive care for several months after birth due to respiratory problems, showed severe developmental delay, microcephaly, sensorineural deafness, hyperactivity, and multiple broncho-obstructive episodes. Also congenital hypothyroidism, low set ears and retrognathia were observed. WABS04 (female) is the first child of Dutch parents, initially diagnosed with FA with unknown genetic cause[45]. She was born at the 7th month of pregnancy weighing 750 g and had epileptic episodes at the age of three, and childhood hyperactivity as described by the mother. At the age of 45, the following clinical features were recorded: growth and mental retardation, deafness, microcephaly, skin pigmentation (*café-au-lait* spots), facial dysmorphy, bulbous nose, clinodactyly of the 5th fingers, insulin-dependent diabetes mellitus and frequent respiratory and middle-ear infections. No typical indications of anemia or malignancies were observed. She died at the

**Table 1 Clinical features of known WABS patients.**

| | WABS01 (v.d. Lelij et al) | WABS02 | WABS03 | WABS04 | WABS05 | WABS06 | WABS07 | WABS08 | Capo-Chichi et al | | | Bailey et al | Eppley et al | | Alkhunazi et al | | | | | Bottega et al | | Rabin et al | |
|---|---|---|---|---|---|---|---|---|---|---|---|---|---|---|---|---|---|---|---|---|---|---|---|
| Gender | M | M | M | F | M | F | M | M | F | M | F | F | F | F | M | M | F | M | M | F | F | M | F |
| Consanguinity | | | | | | | | | | | | | | | | | | | | | | | |
| Microcephaly | | | | | | | | | | | | | | | | | | | | | | | |
| Deafness: Sensorineural | | | | | | | | | | | | | | | | | | | | | | | |
| Prenatal growth restriction | | | | | | | PA | PA | | | | | | | | | | | | | | PA | |
| Postnatal growth restriction | | | | | | | | | | | | | | | | | | | | | | | |
| Intellectual disability | | | | | | | | | | | | | | | | | | | | | | | |
| Retrognathia | | | | | | | | | | | | | | | | | | | | | | | |
| Sloping/small forehead | | | | | | | | | | | | | | | | | | | | | | | |
| Skin abnormalities | | | | | | | | | | | | | | | | | | | | | | | |
| Clinodactily | | | | | | | | | | | | | | | | | | | | | | | |
| Bulbous nose | | | | | | | | | | | | | | | | | | | | | | | |
| Cochlear abnormalities | | | | | | | | | | | | | | | | | | | | | | | |
| Epicanthus | | | | | | | | | | | | | | | | | | | | | | | |
| Small nares | | | | | | | | | | | | | | | | | | | | | | | |
| Small/dysplastic ears | | | | | | | | | | | | | | | | | | | | | | | |
| Brain abnormalities | | | | | | | | | | | | | | | | | | | | | | | |
| Skeletal abnormalities[a] | | | | | | | | | | | | | ST | SF ST | | | TEV | CS | | BSB | | BLE | DE |
| Congenital hypothyroidism | | | | | | | | | | | | | | | | | | | | | | | |
| Short neck | | | | | | | | | | | | | | | | | | | | | | | |
| Seizures (epilepsy) | | | | | | | | | | | | | | | | | | | | | | | |
| Single palmar crease | | | | | | | | | | | | | | | | | | | | | | | |
| Family history malignancy | | | | | | | | | | | | | | | | | | | | | | | |
| Family history miscarriages | | | | | | | | | | | | | | | | | | | | | | | |
| Syndactyly | | | | | | | | | | | | | | | | | | | | | | | |
| Heart abnormalities | VSD | | | | | | | | | TF | | PDA | | | | | PDA ASD | VSD | | | | | SM |
| Hypotonia | | | | | | | | | | | | | | | | | | | | | | | |
| High-arched palate | | | | | | | | | | | | | | | | | | | | | | NP | |
| Lung abnormalities | | | BE | | | | PH | | | | | | | | | | | | | | | | |
| Feeding problems | | | | | | | | | | | | | | | | | | | | | | | |
| Diabetes mellitus | | | | | | | | | | | | | | | | | | | | | | | |
| Kidney abnormalities | | | | | | | KD | | | | | MK | | | | | | | | | | | |
| Hypotelorism | | | | | | | | | | | | | | | | | | | | | | | |

A red color indicates that this condition was described for the particular patient; green indicates that the condition was not found; white indicates unknown.
*PA* placenta abnormality, *ST* small thumbs, *SF* small fibulae, *TEV* talipes equino varus, *CS* craniosynostosis, *BSB* bilateral shortening first metacarpal bone, *BLE* bilateral limitation of extension of the elbow, *DE* dislocation of elbow, *VSD* ventricular septal defect, *TF* tetralogy of fallot, *PDA* patent ductus arteriosus, *ASD* atrial septal defect, *SM* systolic murmur, *NP* narrow palate, *BE* broncho-obstructive episodes, *PH* pulmonary hypoplasia, *KD* kidney dysplasia, *MK* multicystic kidney.
[a] skeletal abnormalities other than fingers or toes.

age of 64, no autopsy report is available. WABS04 had three unaffected siblings and a brother with clinical features that may have been overlapping. The brother was likely affected, too, but he died of heart failure at the age of 50 before a WABS diagnosis was confirmed[45]. WABS05 (male) is the fourth child of non-consanguineous parents from Croatia. He showed prenatal growth retardation; his birth weight, after 36 weeks, was 1660 g. He suffered epileptic seizures at the age of seven and displayed brachy-microcephaly, moderate to severe intellectual disability, bronchial asthma, clinodactyly of the 5th fingers, flexion contractures of thumbs and sandal gap of toes and he is deaf-mute. Cytogenetic investigation showed a 47XXY karyotype and cohesion defects. WABS06 (female) is the older sister of WABS05. Her birth weight, after 37 weeks, was 2100 g. She shows brachy-microcephaly, abnormal skin pigmentation (*café-au-lait* spots), clinodactyly of the 5th fingers, sandal gap of toe and is deaf-mute. At early age, her intellectual development was estimated to be normal but later declined. WABS07 (male) is a fetus of Dutch parents. The pregnancy was prematurely abrogated due to severe growth restriction of the fetus and placental abnormalities.

Furthermore, the fetus showed mild dysmorphic characteristics, lung hypoplasia, increased liver-brain ratio, unilateral kidney dysplasia and skin abnormalities. WABS08 (male) is a younger fetus of the same parents, also from a pregnancy prematurely abrogated due to severe embryonic growth retardation. In addition, multiple miscarriages with unknown genotype were reported, possibly relevant in the light of placental abnormalities observed in DDX11 knockout mice[15]. Table 1 summarizes the clinical phenotype of all known WABS cases, including the seven patients of this study. Notably, while growth retardation, microcephaly, clinodactyly, and deafness appear to be distinctive diagnostic features of WABS, significant anemia was not observed, as opposed to the vast majority of FA subtypes. Some clinical overlap with the DNA damage repair deficiency syndrome NBS is also noteworthy (see ref. [44]) but we also found no clear evidence for the severe immunodeficiency linked to NBS. While conclusive assessment of increased cancer risk, typical for NBS or FA, requires further follow up, WABS patients appear to lack childhood malignancies or other signs of hereditary cancer predisposition. Finally, not all of the manifestations are shared

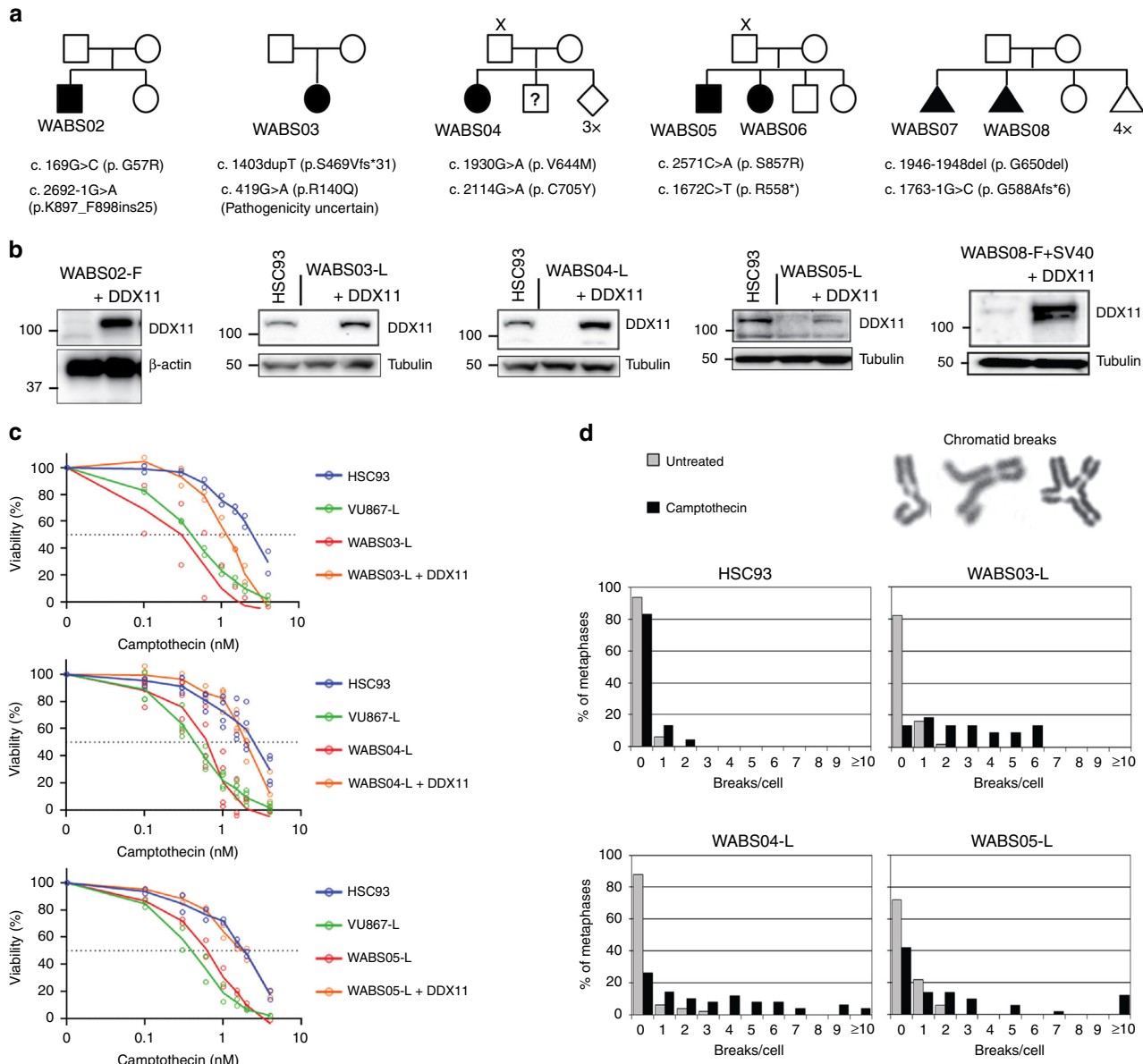

**Fig. 1 Identification of multiple new Warsaw Breakage Syndrome (WABS) cases. a** Pedigrees of seven new WABS patients. X indicates absence of paternal DNA; question mark indicates uncertain whether patient has WABS; diamond indicates unknown sex; triangle indicates fetus. Nomenclature is based on DDX11 transcript variant NM_030653.4. **b** Western blots of patient-derived lymphoblasts (L) and fibroblasts (F). DDX11 protein levels were restored by stable transfection of DDX11 cDNA. Examples of two independent protein analyses are shown. **c** Wild-type lymphoblasts HSC93, FANCM-deficient lymphoblasts VU867-L[80], three WABS-derived lymphoblasts and their complemented counterparts were continuously exposed to increasing Camptothecin concentrations, in two or three independent experiments. After three population doublings of untreated cells, cells were counted and plotted as percentage of untreated cells. **d** metaphase spreads of cells treated as in (**c**) were assessed for chromosome breaks, $n = 50$ for each condition. Depicted examples of counted aberrations include a chromatid gap, a dislocated broken piece and a chromatid interchange figure ('triradial').

between WABS patients, confirming that WABS is a heterogeneous disease sharing overlap with both DNA damage syndromes and cohesinopathies[46–48].

**Molecular characterization of newly identified WABS cases.** Bi-allelic mutations in DDX11 were detected in these patients using whole exome sequencing and/or RNA sequencing, which we confirmed using Sanger sequencing of genomic DNA and cDNA (Fig. 1a and Supplementary Fig. 1). These mutations further extend the list of reported WABS variants (Table 2). For patient WABS03, only one correctly segregating missense

mutation (c.419G>A; p.R140Q) could be identified next to the indicated c.1403 duplication. However, it remains unclear whether DDX11 R140Q is a penetrant pathogenic allele (see below).

Accurate WABS diagnosis depends on DDX11 sequencing which is sometimes hampered by the pseudogene DDX12p, which shares 98% sequence similarity to DDX11, as well as by multiple sub-telomeric DDX11L sequences highly similar to the DDX11 C-terminus[49]. Since DDX12p is also transcribed into mRNA, in several occasions cloning of PCR fragments and subsequent sequencing was required to confirm that the mutations are present in *DDX11* (Supplementary Fig. 1a–e).

**Table 2 Currently reported DDX11 variants in WABS patients.**

| | | Variant 1 | | Variant 2 | |
|---|---|---|---|---|---|
| | | DNA | Protein | DNA | Protein |
| v.d. Lelij et al.[1] | WABS01 | c.2271+2T>C | p.C754P fs*9 | c.2689-2691del | p.K897del |
| Capo-Chichi et al.[6] | 3 patients | c.788G>A | p.R263Q | c.788G>A | p.R263Q |
| Bailey et al.[5] | 1 patient | c.638+1G>A | splice site | c.1888delC | p.R630F fs*23 |
| Eppley et al.[7] | 2 patients | c.1523T>G | p.L508R | c.1949-1G>A | splice site |
| Alkhunazi et al.[4] | Patient 1 | c.606delC | p.Y202* | c.2372G>A | p.R791Q |
| | Patient 2 | c.1133G>C | p.R378P | c.1133G>C | p.R378P |
| | Patient 3+4 | c.2576T>G | p.V859G | c.2576T>G | p.V859G |
| | Patient 5 | c.2638dupG | p.A880G fs*94 | c.2638dupG | p.A880G fs*94 |
| Bottega et al.[8] | 2 patients | c.2507T>C | p.L836P | c.907-920del | p.K303E fs*22 |
| Rabin et al.[9] | 2 patients | c.1763-1G>C | splice site | c.1763-1G>C | splice site |
| Present study | WABS02 | c.169G>C | p.G57R | c.2692-1G>A | p.K897_F898ins25 |
| | WABS03 | c.1403dupT | p.S469V fs*31 | c.419G>A | p.R140Q |
| | WABS04 | c.1930G>A | p.V644M | c.2114G>A | p.C705Y |
| | WABS05+WABS06 | c.2571C>A | p.S857R | c.1672C>T | p.R558* |
| | WABS07+WABS08 | c.1946-1948del | p.G650del | c.1763-1G>C | p.G588A fs*6 |

Note that the pathogenicity of variant c.419G>A (p.R140Q) in patient WABS03 is uncertain.

As reported previously for other WABS cases, we observed reduced DDX11 protein levels (Fig. 1b) and increased CPT sensitivity (Fig. 1c), which correlated with induction of chromosomal breaks (Fig. 1d). We also observed the spontaneous cohesion loss typical for WABS, which was aggravated by CPT treatment and corrected by stably expressing DDX11 cDNA (Fig. 2a). Furthermore, consolidating our earlier observation in two WABS-derived lymphoblast cell lines[50], all WABS cell lines demonstrated high sensitivity to PARP inhibition (PARPi). WABS cells were considerably more sensitive to PARPi than Roberts Syndrome (RBS) and Cornelia de Lange Syndrome (CdLS) cells but similarly sensitive as FA cells (Fig. 2b). In line, WABS01 fibroblasts exhibited increased cohesion loss, G2/M induction and cell death upon treatment with the PARPi Talazoparib (Supplementary Fig. 2).

DDX11 has a role in averting DNA damage in unperturbed cells as well, reflected by the elevated number of γH2AX foci in WABS cells as compared to corrected cells (Fig. 2c). Since DDX11 interacts with DNA replication regulators, its primal activity may reside at the replication fork. Reduced replication fork speed was reported in HeLa cells when DDX11 knockdown was combined with the nucleotide-depleting agent hydroxyurea[40], potentially reflecting the requirement for DDX11 to allow fork progression through difficult-to-replicate regions of DNA. In agreement with this observation, we recently reported slower fork progression upon DDX11 depletion in RPE1 cells[51]. However, in unchallenged and MMS treated chicken DT40 cells, DNA replication appeared to be independent of DDX11[52]. Notably, avian cells may possess increased DNA repair efficiencies which could compensate for DDX11 loss and might explain the reduced impact. To assess replication fork dynamics in WABS lymphoblasts, we performed DNA fiber assays. This revealed a subtle, but consistent decrease of replication fork speed (Fig. 2d). In conclusion, WABS patient cells display CPT-induced chromosomal breaks, as well as spontaneous DNA damage signaling, DNA replication stress and cohesion loss.

**DDX11 missense alleles reduce protein stability.** Remarkably, all patients have at least one hypomorphic variant (missense variant or in frame indel), which may generate a partially active protein. Indeed, we detected faint but specific DDX11 protein bands on western blot (Fig. 3a). Whereas DDX11 mRNA levels of the different WABS cells were all largely comparable to controls

(Fig. 3b), treatment with the proteasome inhibitor marizomib partially restored DDX11 protein levels (Fig. 3c), indicating that DDX11 protein stability is typically reduced in WABS cells.

To test whether this is caused by the identified missense mutations, we overexpressed cDNAs representing the newly identified weak DDX11 alleles in WABS01 fibroblasts (Fig. 3d). In agreement with the unaffected mRNA levels in WABS cells, the mutant DDX11 cDNAs exhibited relatively normal mRNA stability, determined by quantifying the reduction of DDX11 mRNA upon actinomycin-mediated transcription blockage (Supplementary Fig. 3a). We then treated these cells with cycloheximide, which stabilizes mRNA and at the same time blocks translation, to assess protein stability. This revealed that while DDX11 is a rapidly degraded protein already, DDX11 mutants showed further reduced protein stability (Fig. 3e and Supplementary Fig. 3b). The c.419G>A (p.R140Q) variant, found in WABS03, has a minor allele frequency of 0.002 and was predicted to be deleterious by two out of four pathogenicity predicting tools (e.g. Mutation Taster and PolyPhen-2). Overexpression of this variant demonstrated only slightly reduced protein stability. Blocking protein degradation by marizomib or by the lysosome inhibitor chloroquine revealed that the DDX11 mutants are more rapidly stabilized than wild type (wt) DDX11, while R140Q shows a small increase (Fig. 3f, g and Supplementary Fig. 3c, d). So, while we could confirm that different WABS-associated DDX11 missense mutations reduced protein stability, explaining the very low levels of DDX11 protein expression in patient-derived cells, we could only partially confirm that DDX11 R140Q is the disease-causing mutation in the paternal allele of WABS03.

We used the same WABS cell lines to evaluate wt and mutant DDX11 protein localization using immunofluorescence (Fig. 3h). Whereas wt DDX11 shows nuclear localization, several mutants (G57R, S857R, and C705Y) show substantially increased cytoplasmic retention and a more punctuate staining pattern. However, the R140Q mutant localized to the nucleus in a manner similar to wt DDX11.

Despite alterations in their intracellular localizations, the fact that cDNA overexpression of the mutants was able to restore DDX11 protein expression to some extent, suggested that WABS cells from different origins retain residual levels of DDX11 functionality. To further test this, we measured sister chromatid cohesion upon transfection of WABS cells with DDX11 siRNA (Fig. 3i and Supplementary Fig. 4a). Although the extent of

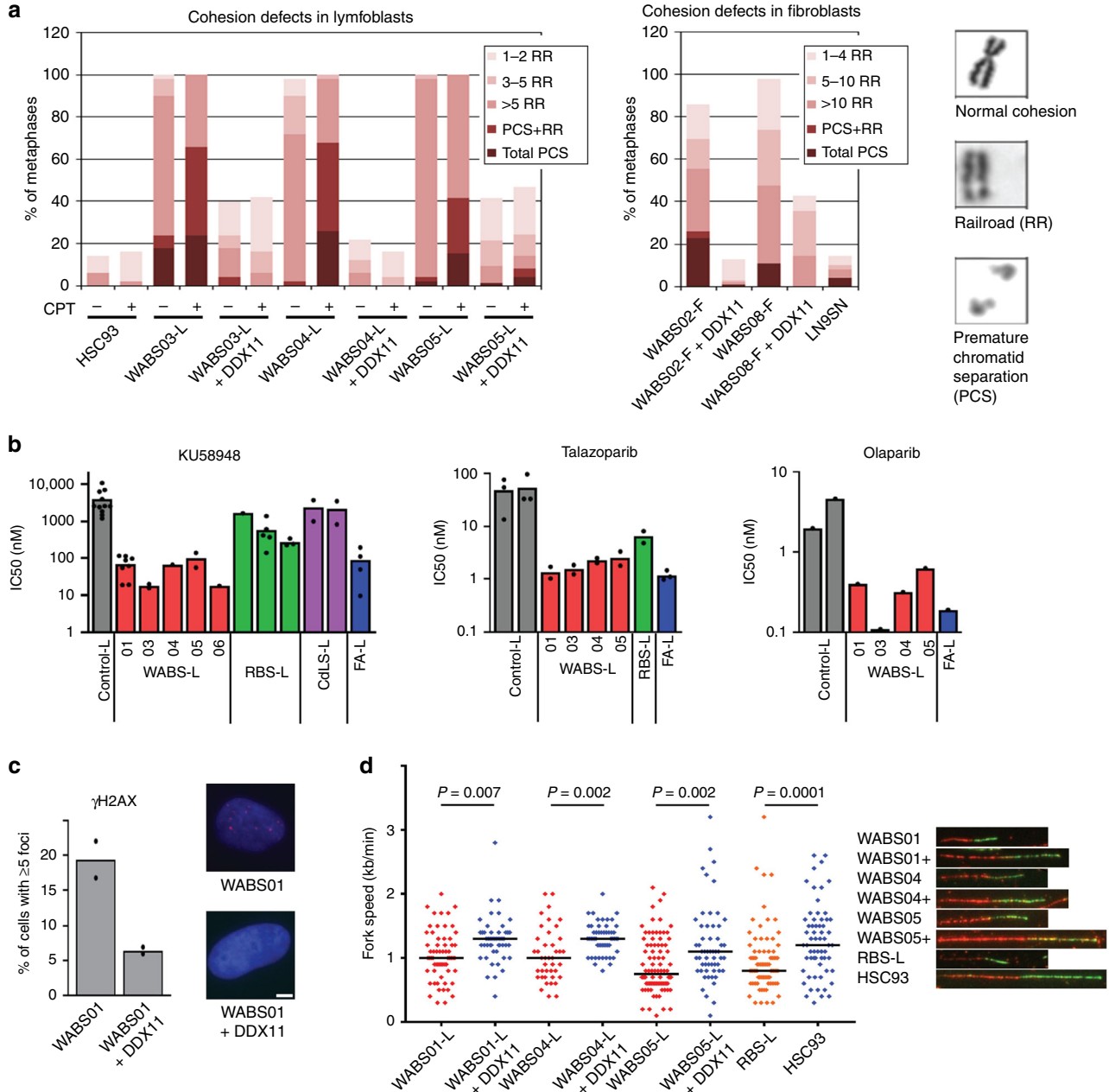

**Fig. 2 WABS cells display cohesion loss, PARPi sensitivity, DNA damage and replication stress. a** Cohesion defect analysis of WABS cells. Per condition, in total 100 metaphases from two independent experiments were assessed. CPT = 2.5 nM Camptothecin treatment for 48 h. Examples of a normal chromosome, a railroad chromosome (RR) and premature chromatid separation (PCS) are shown. **b** Lymphoblasts from different patients were continuously exposed to increasing concentrations of PARP inhibitors KU58948, talazoparib and olaparib. After three population doublings of untreated cells, cells were counted and the amounts were determined as percentage of untreated cells. IC50 values from each dose-response curve were determined using curve fitting. Some KU58948 IC50 values were calculated from previously reported dose-response curves[50]. **c** Immunofluorescence detection of γH2AX foci. n = 100 in two independent experiments. Representative example pictures are shown with DAPI in blue and γH2AX in red. Scale bar, 5 μm. **d** Replication fork speed of WABS lymphoblasts was assessed with a DNA fiber assay using a double labeling protocol. 50 fibers were scored per condition. Example tracks are shown on the right. Blue dots, DDX11 proficient; red dots, DDX11 deficient; orange dots, ESCO2 deficient RBS lymphoblasts (positive control). Black lines indicate the median. P-values were calculated by a non-parametric one-way ANOVA test.

DDX11 mRNA knockdown was modest in some cases, the level of sister chromatid cohesion loss was consistently enhanced by DDX11 siRNA treatment. In summary, multiple patient-derived DDX11 mutations cause destabilization and/or mis-localization of the DDX11 protein. DDX11 is still partially active in the investigated WABS patient-derived cell lines, possibly indicating that complete loss of DDX11 is incompatible with cellular fitness or embryonic survival as is the case in mice[15,16].

**DDX11 loss impairs growth and DDR in a p53-dependent fashion.** We then used CRISPR-Cas9 to delete DDX11 in human diploid RPE1-TERT cells, using previously generated RPE1-TetOn-Cas9 and RPE1-TetOn-Cas9-TP53KO clones[53]. Analysis of multiple DDX11 guide RNA (gRNA) transfected clones revealed only one clone out of eleven with bi-allelic DDX11 inactivation in TP53-wt cells, whereas four out of eight clones in the TP53KO background contained a frame-shift inducing indel

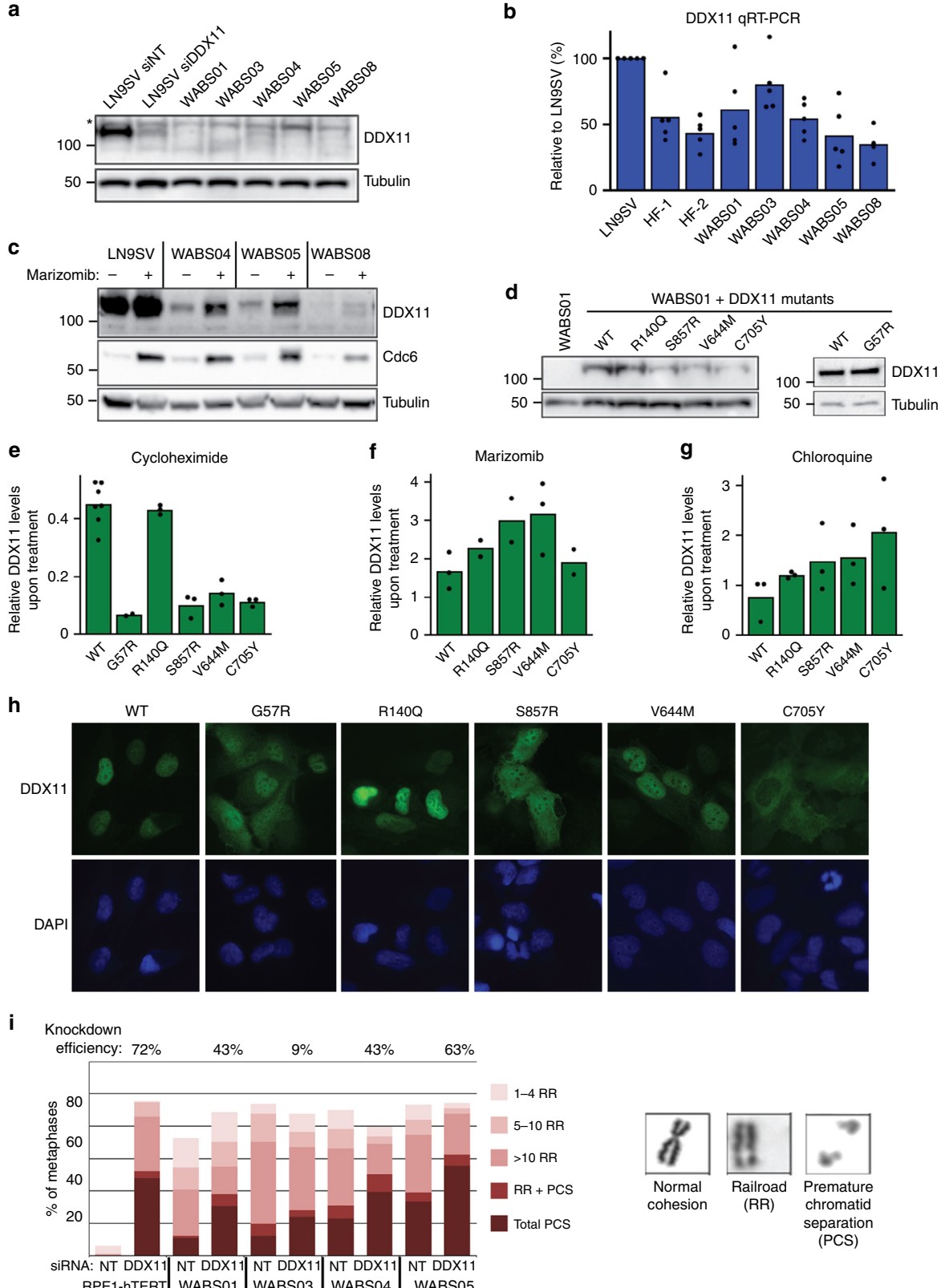

in both alleles (Fig. 4a). This difference might relate to p53-dependent effects on CRISPR efficiency[54,55] but since DDX11KO in wtTP53 cells caused elevated p53 protein levels (Fig. 4b) and reduced growth rate (Fig. 4c), it appears more likely that loss of DDX11 activates p53, hampering survival of DDX11 knockout cell lines. In line with this hypothesis, we

observed increased expression of the p53 target gene p21 both in siDDX11 and in DDX11KO cells (Fig. 4d), and, importantly, transfection with p53 siRNA rescued the proliferation of DDX11KO cells (Fig. 4c). In conclusion, DDX11 inactivation triggers a p53-dependent proliferation delay under normal growth conditions.

**Fig. 3 DDX11 missense alleles reduce protein stability. a** WABS fibroblasts were analyzed for DDX11 protein levels by western blot. As control, LN9SV fibroblasts were transfected with non-targeting or DDX11 siRNA for two days. The asterisk indicates a non-specific band. A representative of three independent experiments is shown. **b** RNA from WABS fibroblasts and three control fibroblasts was analyzed for DDX11 expression by qRT-PCR in five independent experiments. **c** WABS cells were treated with 500 nM marizomib for 5 h to inhibit proteasomal degradation and analyzed by western blot. Cdc6 was included as a positive control. A representative of two independent experiments is shown. **d** WABS01 cells were stably transfected with cDNAs encoding either WT-DDX11 or several patient-derived DDX11 mutants and DDX11 protein levels were analyzed by western blot. A representative of three independent protein analyses is shown. **e** Cells were treated with 62.5 μg/mL cycloheximide for 3 h to inhibit protein synthesis and analyzed by western blot. Bands were quantified using Image Lab software, normalized to tubulin and the decrease of DDX11 protein levels during the cycloheximide treatment was determined for each mutant. **f, g** Similarly, protein degradation was inhibited by treatment with marizomib (500 nM, 5 h) or with the lysosome inhibitor chloroquine (25 μM, 24 h). Increase of DDX11 protein levels during the treatment was determined for each mutant. Examples of western blots that were quantified in (**e**), (**f**), and (**g**) are provided in Supplementary Fig. 3b–d. **h** DDX11 expression and localization in WABS01 cells expressing different DDX11 versions were assessed by immunofluorescence. Two independent experiments were performed, showing comparable results. **i** RPE1-hTERT cells and four WABS fibroblasts were transfected with indicated siRNAs (day 1 and day 4) and analyzed for cohesion defects seven days after the first transfection. Accompanying qRT-PCR (Supplementary Fig. 4a) was performed to determine knockdown efficiency, which is indicated as percentage in the figure. Per condition, in total 150 metaphases from three independent experiments were assessed.

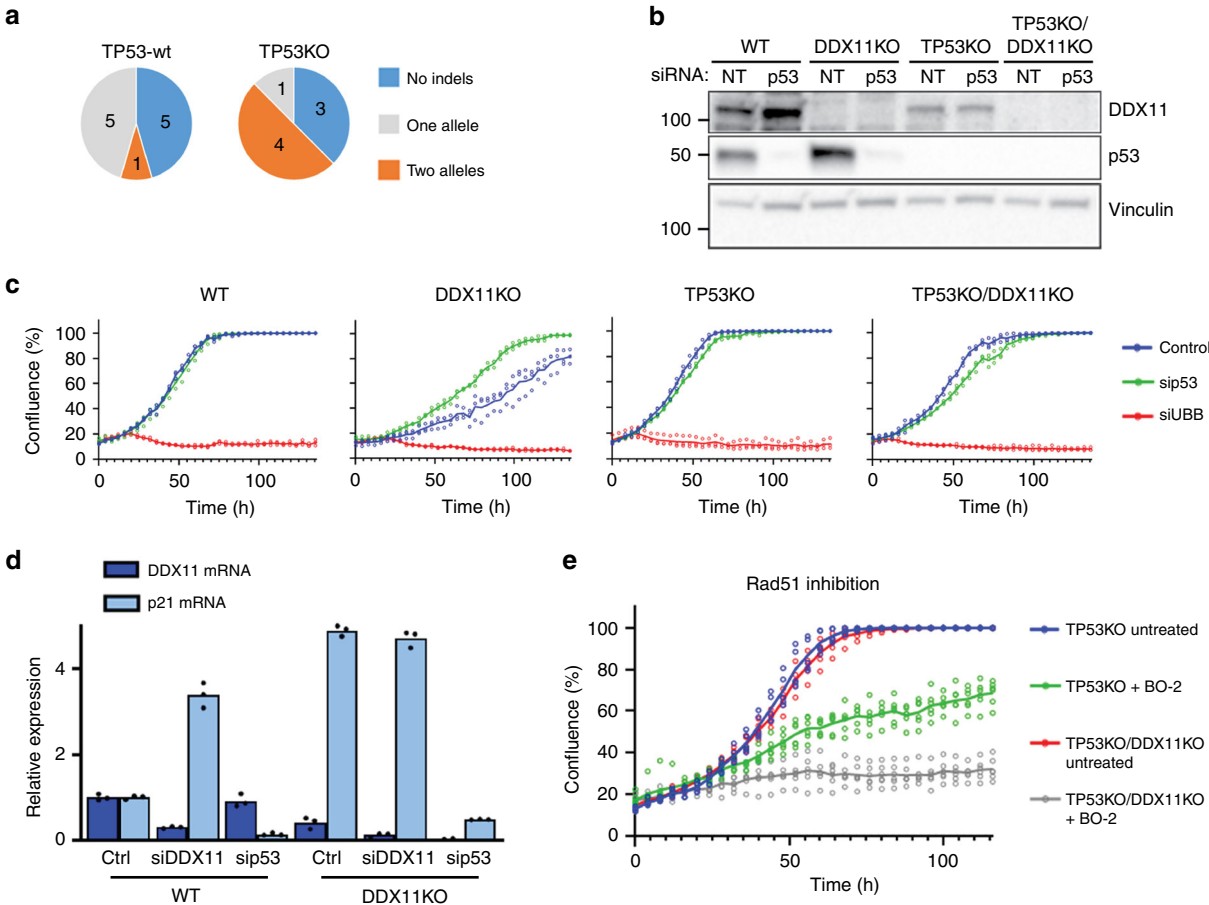

**Fig. 4 DDX11 knockout causes TP53 activation and increased sensitivity to RAD51 inhibition. a** RPE1-hTERT cells and RPE1-hTERT-TP53KO cells, both containing a doxycycline inducible Cas9 construct, were transfected with DDX11 guide RNA and indels were analyzed using Sanger sequencing. More detailed information on guide RNA design and validation of clones is provided in Supplementary Fig. 5. **b** Cells were transfected with indicated siRNA's, lysed after two days and analyzed by western blot. Note that DDX11KO cells have elevated p53 levels. A representative of two independent experiments is shown. **c** In parallel with b, cells were transfected with siRNA and proliferation was monitored using IncuCyte software. UBB siRNA was used to control transfection efficiency. A representative of two independent experiments is shown, with three technical replicates. Note that sip53 specifically accelerates growth of wtTP53-DDX11KO cells. **d** RPE1-hTERT and RPE1-hTERT-DDX11KO cells were transfected with indicated siRNAs. After two days, mRNA levels were assessed with qRT-PCR in three technical replicates. **e** RPE1-hTERT cells were cultured in a 96-wells plate in the presence or absence of the RAD51 inhibitor BO-2 (10 μM). Growth was monitored using IncuCyte software. In total six replicates from two independent experiments are shown.

Similar to WABS cells, RPE1-DDX11KO cells showed increased sensitivity to CPT, as well as to the DNA polymerase inhibitor aphidicolin and PARPi talazoparib (Supplementary Fig. 5a). Moreover, analysis of 53BP1 foci and γH2AX foci revealed increased DNA damage signaling (Supplementary Fig. 5b) and we detected an increased number of asymmetric forks (Supplementary Fig. 5c). Finally, DDX11 deficient cells exhibited increased sensitivity to the RAD51 inhibitor BO-2

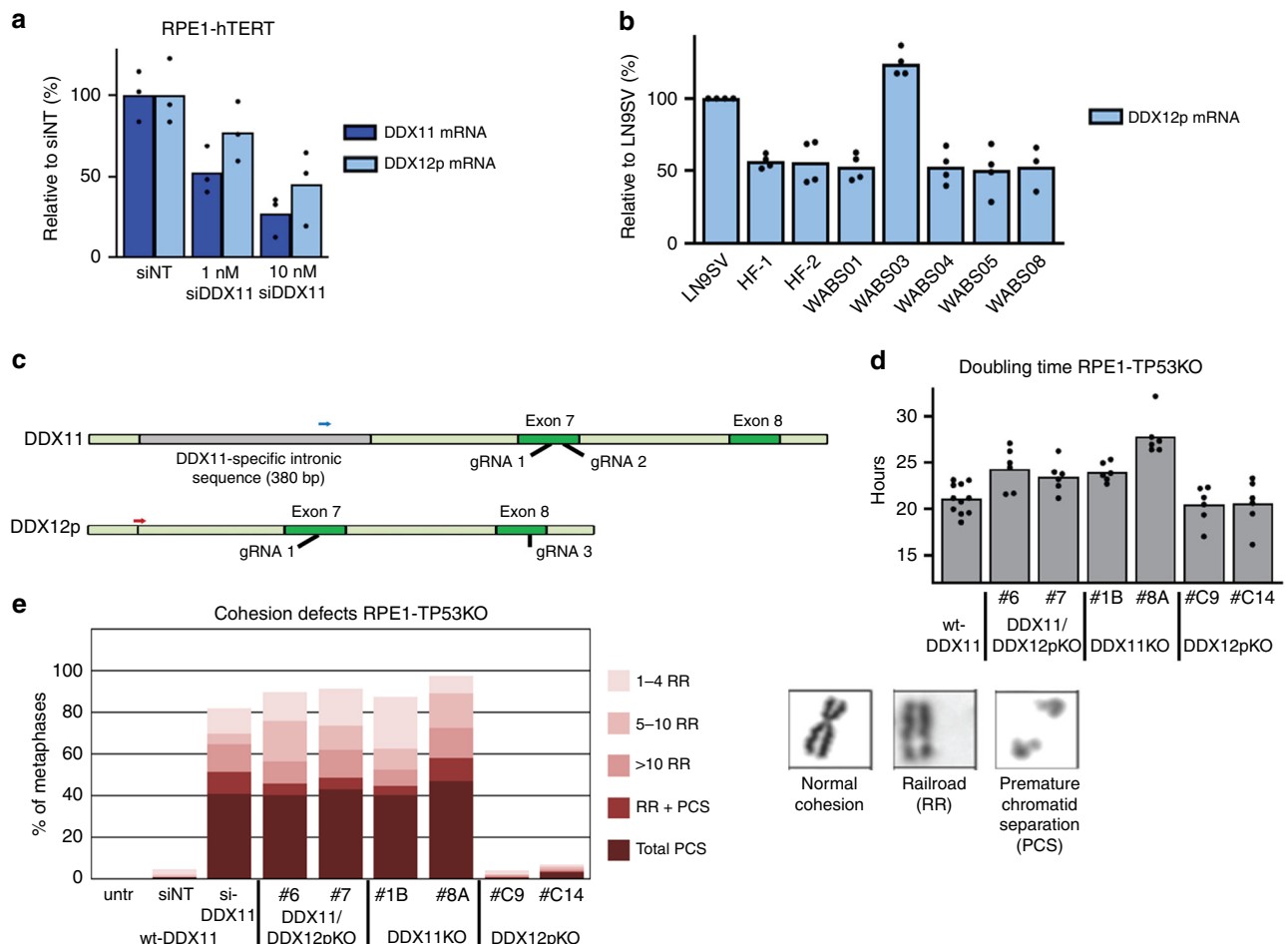

**Fig. 5 No evidence for a redundant role of the pseudogene DDX12p in cohesion and proliferation. a** RPE1-hTERT cells were transfected with siRNA to DDX11 (1 nM or 10 nM) and analyzed for DDX11 and DDX12p mRNA levels using specific qRT-PCR. The specificity of qRT-PCR primers was validated in Supplementary Fig. 5. **b** DDX12p mRNA levels were assessed in three SV40 transformed control fibroblasts and five WABS fibroblasts. **c** CRISPR design for constructing DDX11 and DDX12p knockouts in RPE1-hTERT-TP53KO cells. For more detailed information and validation of clones, see Supplementary Fig. 6. **d** A panel of RPE1-hTERT-TP53KO cells containing specific DDX11 and/or DDX12p knockout was seeded in 96-wells plates and growth rate was analyzed using IncuCyte software. The resulting growth curves were used to calculate doubling times. **e** The same panel was analyzed for cohesion defects. As control, RPE1-hTERT-TP53KO cells were transfected with siDDX11 for two days. Per condition, in total 100 metaphases from two independent experiments were assessed.

(Fig. 4e), indicating that proliferation of DDX11 deficient cells is highly dependent on continuous DNA damage repair via a RAD51-dependent pathway.

**No redundant role of DDX12p in cohesion and proliferation.** Considering high sequence similarities, we aimed to investigate whether the pseudogene DDX12p, which is absent in mice, could partially compensate for loss of DDX11. We designed qRT-PCR primers that specifically amplify DDX11 or DDX12p, utilizing a small difference in the sequences of exon 17 of DDX11 and DDX12p, respectively (Supplementary Fig. 4b). This revealed that the DDX11 siRNA used here also depletes DDX12p mRNA (Fig. 5a), providing a possible explanation for increased cohesion loss in WABS cells upon treatment with this siRNA. Interestingly, WABS cells express DDX12p mRNA at comparable levels as control cells (Fig. 5b). We constructed specific DDX11 or DDX12p knockout cells, using gRNA target sites close to an intronic region in DDX11 which is absent in DDX12p (Fig. 5c). This enables specific amplification of both DDX11 and DDX12p genomic DNA by specific forward primers, either inside (DDX11, blue arrow) or spanning this region (DDX12p, red arrow). We generated a panel of six cell lines (two clones for each gRNA) that

is described in detail in Supplementary Fig. 6. Importantly, we found that loss of DDX11, but not of DDX12p, impairs cellular proliferation (Fig. 5d) and induces cohesion loss (Fig. 5e) in RPE1 cells. To determine the full length human DDX12p mRNA sequence, unavailable through validated genomic databases, we cloned and sequenced multiple PCR fragments (Supplementary Fig. 7a) and also performed a 3'RACE PCR (Supplementary Fig. 7b). This revealed that an intact start codon, all exons and a poly-A tail can be found in DDX12p transcripts, although some exon-skipping isoforms were also detected. Apparently an intact DDX12p ORF might exist. However, we also found a 5 bp deletion in exon 8 leading to a predicted DDX12p protein of only 300 amino acids, lacking large parts of the putative helicase domain of DDX11 yet containing the region homologous to the Timeless binding domain of DDX11 (186–232)[41] (Supplementary Fig. 7a). In conclusion, while DDX12p encodes an RNA product that is processed by RNA splicing and poly-adenylation, we found no evidence that DDX12p (m)RNA encodes a protein that functionally compensates for DDX11 helicase loss.

**DDX11 deficient cells do not tolerate G4 stabilization.** Anti-parallel G-quadruplex (G4) structures, four-stranded structures

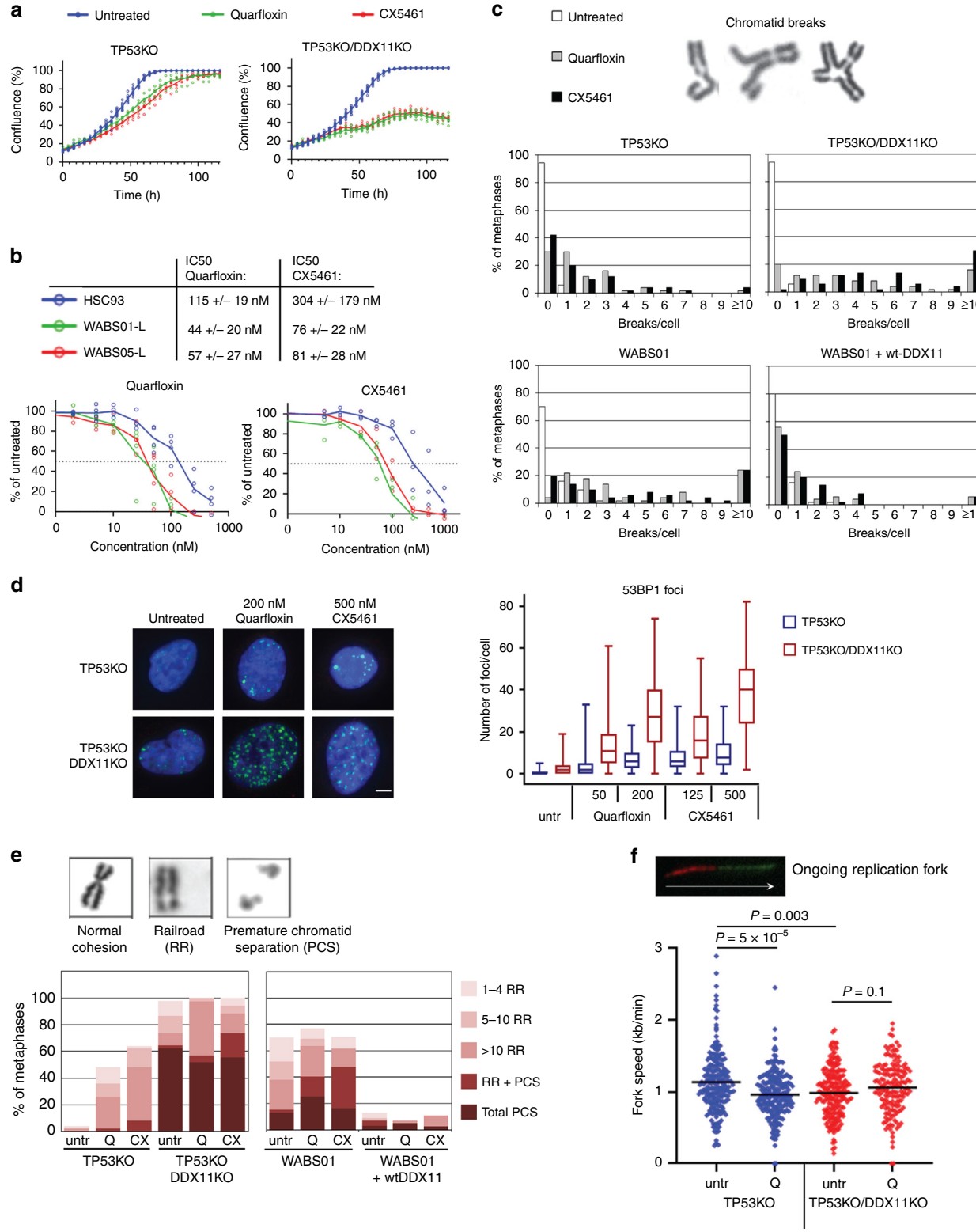

formed by guanine-rich nucleic acids, were previously reported to be a particular DDX11 substrate in vitro[22]. Therefore, we investigated the effects of two different G4-stabilizing compounds, quarfloxin (CX3543) and CX5461[56]. In both RPE1-DDX11KO cells and in WABS cells, G4 stabilization significantly inhibited proliferation (Fig. 6a, b and Supplementary Fig. 8a) and caused the accumulation of chromosomal breaks (Fig. 6c). We also observed increased DNA damage signaling, indicated by 53BP1

foci (Fig. 6d). Strikingly, G4 stabilization was sufficient to induce cohesion loss in RPE1 cells and aggravated cohesion defects in WABS cells (Fig. 6e). WABS01 + wtDDX11 cells did not show cohesion loss upon G4 stabilization, suggesting that overexpression of DDX11 confers cellular resistance to these compounds. Lastly, we assessed replication fork dynamics by a DNA fiber assay. Quarfloxin had no clear effect on origin firing (Supplementary Fig. 9a), but we observed reduced fork speed (Fig. 6f)

**Fig. 6 DDX11 confers resistance to G-quadruplex stabilization. a** Cells were cultured in 96-wells plates and treated with 200 nM quarfloxin or 500 nM CX5461. Growth was monitored using IncuCyte software in three technical replicates. **b** WABS01, WABS05 and HSC93 control lymphoblasts were continuously exposed to increasing concentrations quarfloxin or CX5461. After three population doublings of untreated cells, cells were counted and plotted as percentage of untreated cells. Data from three independent experiments are shown. IC50 values from each dose-response curve were determined using curve fitting and shown as averages ± standard deviations. **c** Cells were treated with 200 nM quarfloxin or 500 nM CX5461 for 24 h and chromosomal breaks were scored in 50 metaphase spreads per condition. **d** Cells were treated as indicated and 53BP1 staining was analyzed by immunofluorescence. In total at least 85 cells per condition were counted for two independent experiments. Boxes extend from the 25th to the 75th percentile, whiskers from the smallest to the largest value, lines indicate the median. **e** Cells were treated with 200 nM quarfloxin or 500 nM CX5461 for 24 h and cohesion defects were analyzed in 50 metaphases per condition. **f** Cells were treated with 200 nM quarfloxin (Q) for 24 h and assessed with a DNA fiber assay using a double labeling protocol. In total at least 165 fibers were scored per condition in two independent experiments. The example track represents an ongoing fork. Black lines indicate the mean. P-values were calculated by a non-parametric one-way ANOVA test.

and an increased amount of asymmetric fibers (Supplementary Fig. 9b), suggesting increased stalling of DNA replication forks.

Since both quarfloxin and CX5461 may inhibit ribosomal DNA (rDNA) transcription[57,58], we asked whether repressed ribosomal gene expression as such contributes to the observed toxicity in WABS cells. To this end, we assessed the effect of the rDNA binding agent Bmh21, which causes blockage of RNA polymerase I and degradation of its catalytic subunit[59,60]. At concentrations that clearly block rDNA transcription (Supplementary Fig. 8b), WABS01 cells were not sensitive to Bmh21 (Supplementary Fig. 8c). Therefore, we conclude that the sensitivity of DDX11 deficient cells to quarfloxin and CX5461 primarily relates to stabilization of G4s or similar DNA structures by these drugs. Together, our findings suggest that unresolved, putative DDX11 substrates by themselves are a source of both DNA damage and sister chromatid cohesion loss, possibly at DNA replication forks.

**Distinct roles of FANCJ and DDX11 in G4 stabilizer responses**. We wondered how the role of DDX11 in G4 stabilizer protection compared to the related Fe-S helicase FANCJ. We generated FANCJ knockouts in RPE1-TP53KO and in RPE1-TP53KO-DDX11KO cells (Fig. 7a). As reported previously[52], double DDX11 FANCJ knockout resulted in a synergistic growth defect even in the absence of genotoxic agents (Fig. 7b), suggesting roles for DDX11 and FANCJ in normal proliferation. FANCJKO had a larger effect on MMC response than DDX11KO, both in recovery from transient exposure (Fig. 7c) and during continuous exposure (Fig. 7d). Combined depletion further enhanced MMC sensitivity, indicating that FANCJ and DDX11 function in separate repair pathways, as previously hypothesized[52].

FANCJKO cells also displayed increased sensitivity for the small-molecule G4 stabilizer pyridostatin, albeit less than DDX11KO cells. Surprisingly, treatment with quarfloxin and CX5461 hardly affected FANCJKO cells (Fig. 7c, d). A comparison of patient-derived fibroblasts showed high MMC sensitivity in FANCJ patient cells, whereas WABS cells were most sensitive to quarfloxin and CX5461 (Supplementary Fig. 10). The observed drug sensitivities of FANCJKO and DDX11KO cells coincided with DNA damage signaling, as detected by γH2AX and TP53BP1 *focus* formation (Fig. 7e, f). In addition, DDX11KO cells exhibited increased DNA damage signaling in untreated conditions, which was not detected in FANCJKO cells but was further exacerbated in double knockout cells. This suggests that FANCJ functions in a different pathway as DDX11 to mitigate DNA damage arising from endogenous sources.

We further investigated FANCJ- and DDX11-dependent drug responses by transient crRNA transfection in Cas9 expressing cells. This approach was validated with a lethal crRNA, which efficiently blocked cell growth (Supplementary Fig. 11a), and by indel analysis (Supplementary Fig. 11b). As expected, acute DDX11 knockout sensitized to quarfloxin, CX5461, pyridostatin

and MMC, whereas FANCJ depletion sensitized to MMC and possibly also to pyridostatin (Supplementary Fig. 11c, d). However, no effect of FANCJ knockout on quarfloxin or CX5461 sensitivity was observed. Targeting two other putative G4 helicases, the RecQ family members BLM and WRN[61], revealed no effect of WRN, whereas targeting BLM mildly sensitized to all compounds and resulted in additive sensitivity in DDX11KO cells (Supplementary Fig. 11c–e). Together, we conclude that, of the disease-linked DNA helicases tested here, DDX11 is the dominant DNA helicase resolving substrates that are targeted by the G4 stabilizers quarfloxin and CX5461.

**Sister chromatid cohesion requires DDX11 helicase activity**. To further investigate the role of DDX11 helicase activity in sister chromatid cohesion, we introduced DDX11-K50R, a previously described DNA helicase-dead mutant[22], into WABS01 cells and TP53KO-DDX11KO RPE1 cells (Fig. 8a). Whereas wtDDX11 overexpression rescued sister chromatid cohesion loss in DDX11 deficient cells, overexpression of DDX11-K50R had no protective effect (Fig. 8b). Furthermore, DDX11-K50R could not restore DNA replication fork speed (Fig. 8c) or resistance to multiple G4 stabilizers (Supplementary Fig. 12a), talazoparib and CPT (Supplementary Fig. 12b) in DDX11KO RPE1 cells. These findings indicate that the helicase activity is essential for DDX11 function.

The observation that WABS cells possess low expression of DDX11 and residual DDX11 activity predicts that some patient-derived DDX11 missense alleles are hypomorphic but still functional when overexpressed. Indeed, we found that three patient-derived missense mutants could at least partially correct the cohesion loss in WABS01 cells: R140Q, V644M, and S857R (Fig. 8d). Two others, G57R and C705Y, showed no activity, indicating these are null-alleles. The C705Y allele originates from WABS04, indicating that the other allele, DDX11 V644M is hypomorphic. The G57R allele is derived from patient WABS02. The second DDX11 allele of WABS02 harbors a splice-site mutation resulting in retention of intron 26 and an in-frame insertion of 25 amino acids at DDX11 C-terminus, which may yield a partially functional DDX11 protein. We noticed that G57R is located close to the Walker A helicase motif I, whereas C705Y is located in helicase motif IV (Fig. 8e). Both regions are highly conserved between different DDX11 orthologues and, to a lesser extent, between other FeS-cluster containing SF2 helicases. This suggests that DNA helicase activity is also impaired in these mutants. Indeed, DDX11-C705Y showed impaired in vitro helicase activity, while DDX11-R140Q acted similarly as WT (Fig. 8f). Moreover, the quarfloxin sensitivity of WABS01 fibroblasts could be rescued by significantly overexpressing patient-derived hypomorphic point mutants but not by helicase-dead DDX11 point mutants (Supplementary Fig. 12c). To prove the essential role of the DDX11 helicase domain in vivo, we generated DDX11^{G57R/WT} mice by oligonucleotide-directed

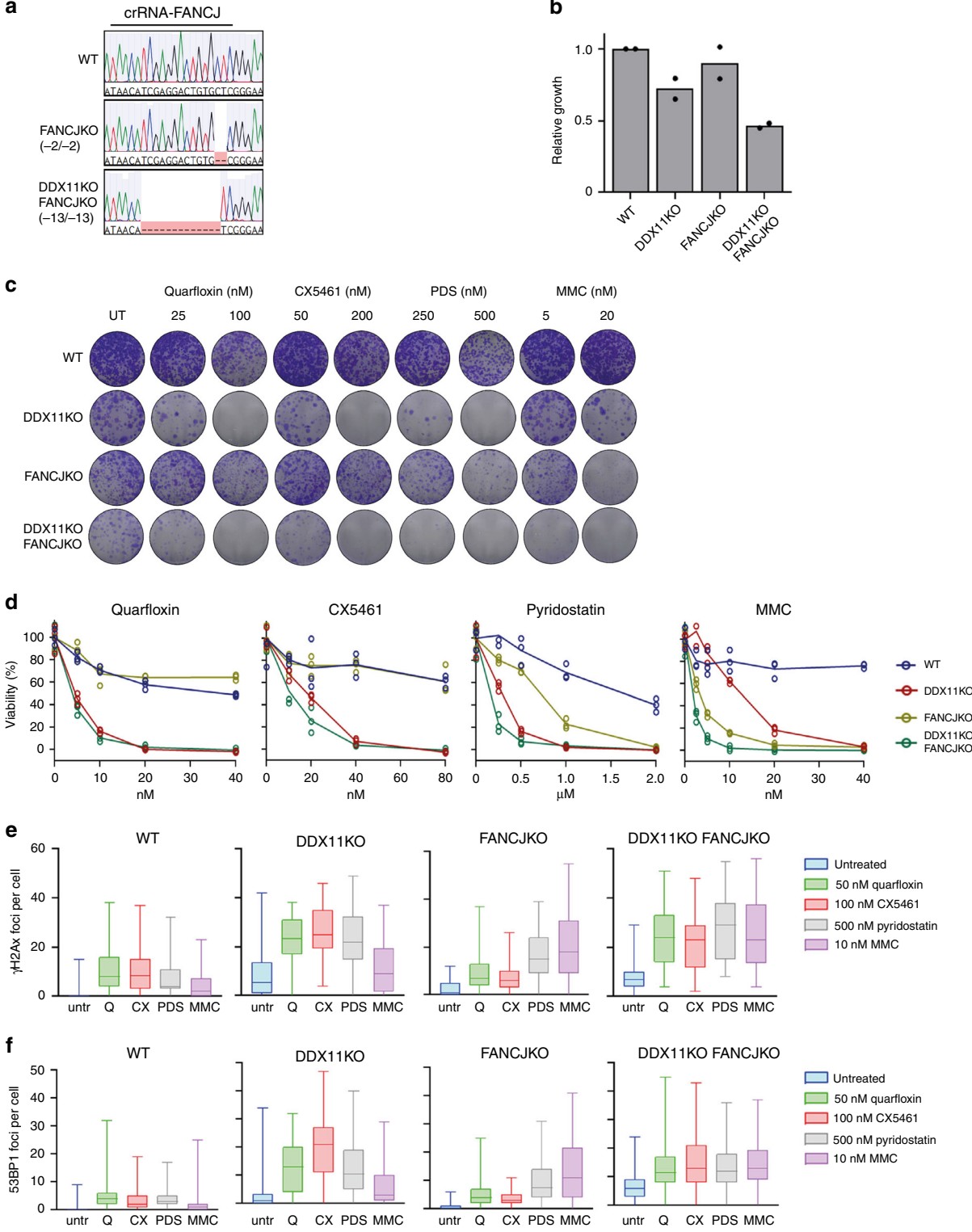

**Fig. 7 Distinct functions of FANCJ and DDX11 in response to G4-stabilization. a** Alignment of sequences of FANCJKO and DDX11KO-FANCJKO at the crFANCJ target site. **b** Cell growth relative to RPE1-TP53KO cells, as assessed by counting three days after seeding. Bars represent mean of two biological replicates. **c** Clonogenic survival assay was performed in cells treated as indicated for 24 h, and subsequently cultured for nine days. Representative images from two separate experiments are shown. **d** Cell titer blue assessed viability following five days of the indicated treatments. Lines indicate the mean of three technical replicates. **e, f** Counts of γH2AX (**e**) and TP53BP1 (**f**) foci per cell after 24 h treatments (n = 50 cells for two separate experiments). Boxes extend from the 25th to the 75th percentile, whiskers from the smallest to the largest value, lines indicate the median.

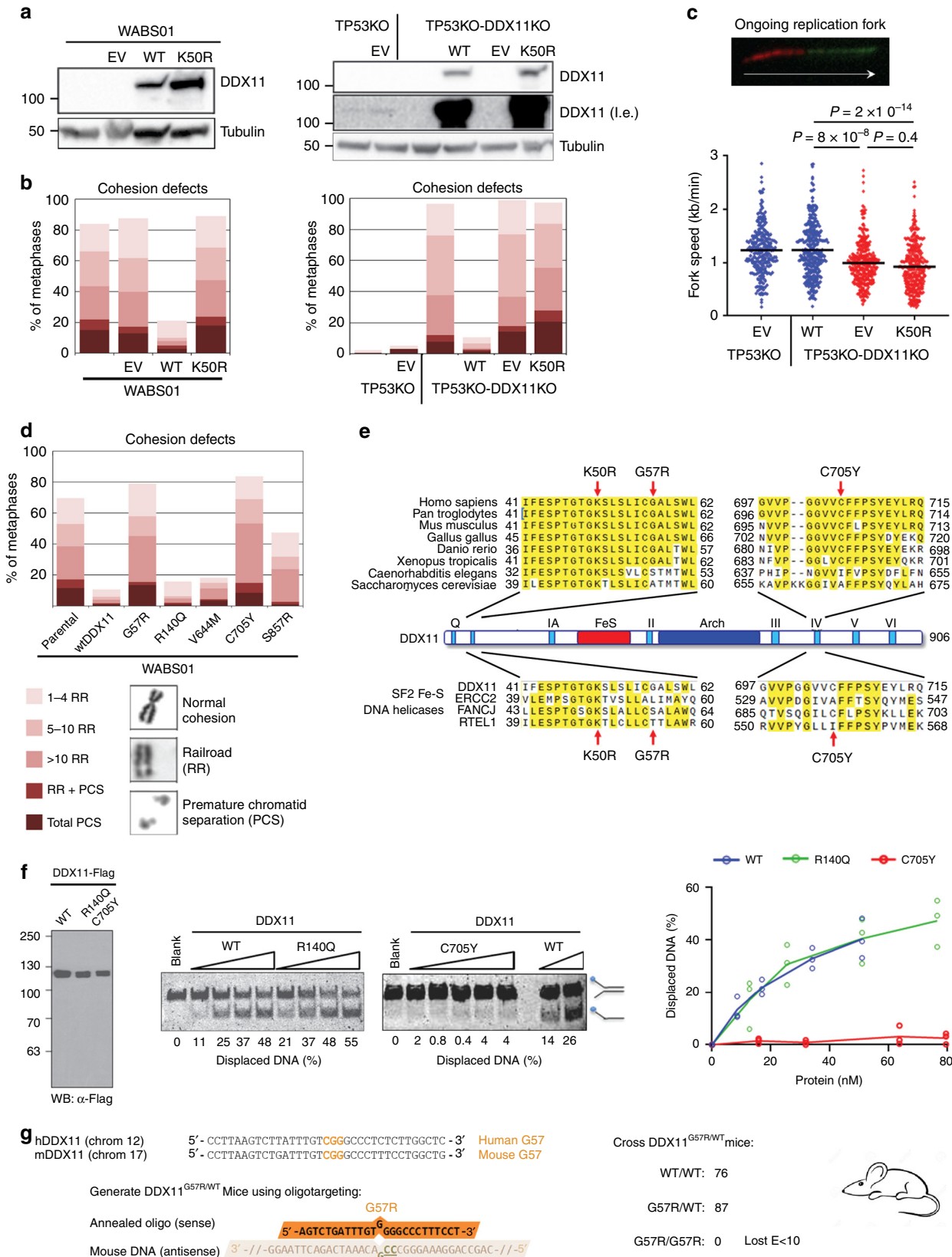

gene modification (Fig. 8g). Intercrossing of these mice did not yield any homozygous DDX11^{G57R/G57R} offspring among 87 heterozygous and 78 wild-type animals. Genotyping of prenatal stages revealed lethality before day 10 of embryonic development, indicating that the presumed helicase-dead DDX11-G57R is a

*null* allele (Fig. 8g). Taken together, we conclude that the DNA helicase domain of DDX11 facilitates normal DNA replication, to prevent DNA damage and cohesion loss, presumably by resolving G4 structures that arise when replication forks travel through G-rich regions.

**Fig. 8 DDX11 helicase activity supports cohesion, DNA replication and embryonic survival. a** WABS01 cells (left panel) and RPE1-TP53-DDX11KO cells (right panel) were stably transduced with empty vector (EV), wtDDX11 or DDX11-K50R and analyzed by Western blot. l.e. long exposure. A representative of two independent protein analyses is shown. **b** Cells were analyzed for cohesion defects. Per condition, in total 100 metaphases from two independent experiments were assessed. **c** Replication fork speed was assessed with a DNA fiber assay using a double labeling protocol. In total at least 200 fibers were scored per condition in two independent experiments. The example track represents an ongoing fork. Blue dots, DDX11 proficient; red dots, DDX11 deficient. Black lines indicate the mean. *P*-values were calculated by a non-parametric one-way ANOVA test. **d** WABS01 cells, stably transfected with different DDX11 mutants, were analyzed for cohesion defects. Per condition, in total 150 metaphases from three independent experiments were assessed. **e** Alignment of different DDX11 orthologues (top) and related SF2 DNA helicases (bottom). Conserved motifs are indicated. Arrows point at the positions of K50R, G57R, and C705Y mutations. **f** In vitro DNA helicase assay of patient-derived DDX11 mutants. Purified, recombinant Flag-tagged proteins were analyzed by western blot (left). Proteins were incubated at increasing concentrations with a fluorescently labeled forked DNA substrate. Helicase reaction products were resolved by gel electrophoresis. Percentage of displaced DNA was determined using Image-J software. Right panel: quantification from three independent experiments. **g** DDX11-G57R heterozygous mice were generated by oligonucleotide-directed gene modification in mouse embryonic stem cells substituting glycine codon 57 for an arginine codon (C>G substitution). Subsequent intercrossing of heterozygous mice failed to produce homozygous offspring.

## Discussion

We show here that WABS cells possess residual DDX11 activity and find that several patient-derived DDX11 mutants encode a functional but unstable protein. Although most published DDX11 mutants were not functionally studied, most if not all WABS patients appear to harbor at least one pathogenic mutation that could potentially yield a protein product with partial activity. Indeed, DDX11-R378P has reduced protein stability[4], whereas DDX11-R263Q, although located in the FeS cluster and displaying reduced helicase activity, could still unwind a forked duplex structure at higher concentrations[6]. These observations, combined with the embryonic lethality of DDX11 knockout mice[15,16], the lack of functional redundancy of the pseudogene DDX12p and the fact that DDX11 knockout causes severe, p53-dependent growth inhibition in human RPE1 cells, suggest that minimal DDX11 protein expression is essential to support cellular fitness and normal human development. Notably, caution is needed in the diagnosis of disease-causing DDX11 mutations, for instance when prenatal testing for WABS would be intended. Mutations with an intermediate effect on DDX11 protein function are common in WABS patients but are difficult to distinguish from rare, non-pathogenic DDX11 variants. For example, DDX11-R140Q could rescue sister chromatid cohesion in WABS01 cells when overexpressed. It showed reduced protein stability in some experiments but the effect was mild. Nevertheless, patient WABS03 is undoubtedly a WABS patient based on the rescue of cellular defects by the introduction of wild-type DDX11, and we found no other intronic or exonic mutation in the relevant DDX11 allele that correctly segregated with the inheritance of the disease. DDX11 mRNA expression in WABS03 cells was similar to that observed in control cells, providing no evidence for impaired DDX11 gene promoter function.

With the seven cases of this study, a total of 23 WABS cases have now been described. Whereas WABS clinically resembles FA, the anemia phenotype is lacking. WABS also resembles NBS, but evidence for severe immune deficiency and early onset of leukemia or other cancers observed in NBS patients has not been found in WABS so far. The oldest WABS patient reported, WABS04, died at the age of 64 with no evidence of cancer. The clinical spectrum of WABS is heterogeneous and also shows considerable overlap with the cohesinopathies Roberts Syndrome (RBS) and Cornelia de Lange Syndrome (CdLS), with the most notable exceptions being limb reductions, which are not found in WABS (although clinodactyly is observed), and abnormal skin pigmentation including *café-au-lait* spots, which are not found in RBS and CdLS[1,4–9]. WABS has been classified as a cohesinopathy because of the spontaneous railroad chromosomes and PCS that are observed in metaphase spreads. This characteristic is also found in RBS, but not in CdLS, which may be explained by the milder impact of CdLS mutations on cohesin function. In yeast,

the distinct processes regulated by cohesin were shown to be differentially affected by modulating cohesin levels[62]. The clinical overlap between RBS and CdLS could be attributed to overlapping effects of reduced cohesion on gene transcription, whereas differences might relate to additional defects in mitotic sister chromatid cohesion and DNA repair, typical for RBS[63]. Although most cohesinopathy-derived cells display some level of increased genotoxic sensitivity, the MMC-induced breakage phenotype is particularly pronounced in WABS cells, reported in most (except one[4]) published cases and in the three WABS cases in this study in which chromosomal breakage was investigated. Moreover, drug toxicity patterns, in particular the sensitivity to PARPi, indicate either impaired homologous recombination in WABS cells[52] or pinpoint to increased sensitivity to DNA replication stress[64]. In line with these observations, multiple of the clinical manifestations of WABS overlap with those observed in chromosomal instability syndromes such as FA[65] and NBS[66]. Thus, WABS can be classified as both a cohesinopathy and a chromosomal breakage syndrome, even though evidence for increased cancer risk is missing.

Our data show that the sensitivity of DDX11 deficient cells to quarfloxin and CX5461 exceeds that of cells deficient for other putative G4 helicases FANCJ, BLM, and WRN. By contrast, it has been reported that the G4 stabilizer telomestatin (TMS)[67] is toxic to FANCJ deficient cells, whereas DNA damage inflicted by TMS is hardly affected by DDX11[68]. A possible explanation for this difference could be that like G4 helicases, also G4 stabilizers act on a specific subset of G4s, depending on their architecture and/ or subcellular localization[69]. While the exact DNA substrate specificities of quarfloxin are incompletely understood, some observations are noteworthy. Quarfloxin stabilizes different variants of G4 structures[56], but accumulates in the nucleolus in cells and has a preference for ribosomal DNA and promoter sequences[58]. TMS on the other hand stabilizes the formation of telomeric G4 structures, thereby inhibiting telomerase[70]. Interestingly, studies in *S. pombe* suggest that telomeric G4 motifs form intramolecular G4s, whereas in ribosomal DNA predominantly intermolecular G4s are formed[71]. Considering that in vitro experiments revealed that DDX11 preferentially resolves bi-molecular over uni-molecular G4s[22,68] and DDX11 was previously reported to reside in the nucleolus[72], the observed sensitivity of DDX11 deficient cells to quarfloxin may originate from unresolved nucleolar G4s.

Apart from their roles in promoting sister chromatid cohesion and preventing DNA breaks in unperturbed conditions, Chl1 and DDX11 are also involved in the response to DNA replication stress[14,17,25,38,73–75], a function probably executed by promoting RPA loading and Chk1 activation[38,75]. It has been proposed that the DNA helicase activity of yeast Chl1 is only required for replication fork progression and the response to fork stalling,

whereas its role in sister chromatid cohesion was suggested to be helicase-independent but relies on interaction with Ctf4[36]. Importantly, however, we show that the DNA helicase function of human DDX11 is not only required to prevent lethal DNA damage in response to G4 stabilizing drugs but also for sister chromatid cohesion, in line with observations in chicken DT40 cells[43]. Besides the previously characterized helicase-deficient mutant K50R[76], also two patient-derived mutations in or near conserved helicase motifs (G57R and C705Y) impair the rescue of cohesion defects in DDX11 deficient cells. Importantly, heterozygous G57R knock-in mice failed to produce homozygous G57R offspring due to early onset embryonic lethality, further suggesting that G57R DDX11 is a *null* allele and the helicase domain is crucial for development. We thus propose that the requirements of the DDX11 helicase function for both DNA repair and sister chromatid cohesion are inextricably linked.

Why does loss of DDX11 result in cohesion loss? Based on its interaction with Fen1, it has been suggested that Chl1 allows efficient maturation of Okazaki fragments, facilitating the subsequent loading of cohesin as the fork progresses[19,20,37]. Another model proposed that Chl1 supports the ability of cohesin rings, which are already bound to double-stranded DNA, to capture a second ssDNA molecule[77]. DDX11 deficiency might also result in broken replication forks, which require the removal of cohesin for a rapid repair process, at the cost of cohesion loss[53]. Based on the helicase dependency in our experiments and the known interaction with a number of proteins involved in lagging strand replication[20,38,39,41], DDX11 may function primarily to resolve complex secondary structures in the lagging strand. Possibly, the binding of DDX11 to several replisome components, such as Timeless, facilitates the correct positioning of DDX11 to unwind its substrates. In this model, the observed cohesion loss, replication delay, DNA damage and growth inhibition resulting from DDX11 depletion are linked to the same process: failure to resolve secondary DNA structures in the lagging strand, leading to fork stalling and DNA breaks. This may hamper cohesin loading and trigger cohesin removal to facilitate DNA repair. It will be important to further unravel which exact structures are resolved by DDX11 and targeted by drugs such as quarfloxin and CX5461, and the context in which these occur in cells.

## Methods

**Compounds**. The following drugs were used: camptothecin (CPT), actinomycin D, cycloheximide, marizomib, chloroquine, Bmh21, BO-2, (all Sigma-Aldrich), aphidicolin (Santa Cruz), talazoparib (Axon Medchem), quarfloxin (CX-3543, Adooq Bioscience) and CX-5461 (Selleckchem).

**Samples and sequencing**. The research on patient material was carried out after approval by the institutional review board of the VU University Medical Center, adhered to local ethical standards. Appropriate informed consent was obtained from patients or from parents/relatives as applicable. For DNA diagnosis, DNA was extracted from whole blood or EBV-transformed lymphoblastoid cell lines. Genomic DNA was isolated using the QIAmp blood kit (Qiagen). Direct Sanger sequencing of DDX11 coding exons was performed for WABS02 and WABS03. WABS05 was analyzed with a custom Haloplex (Agilent) kit targeting the coding regions of all known cohesinopathy genes using 200 ng of genomic DNA to obtain sequencing libraries. Samples tagged with a unique barcode and pooled before 150 bp pair-ended sequencing on the Illumina Miseq platform. SureCall software (version 2.0, Agilent) was used to detect variants. WABS03 and WABS05 primary fibroblasts were both analyzed by RNA sequencing upon treatment with cycloheximide. The cDNA was synthesized with random hexamers and barcoded DNA adapters were ligated to both ends of the double-stranded cDNA and subjected to PCR amplification. The resultant library was sequenced on an Illumina HiSeq 2000. Sample WABS04 was enriched for the whole exome using Nimblegen Exome V.3 oligo library kit, barcode-tagged and pooled with a total of six samples, and sequenced on one HiSeq 2000 illumina lane using the 100 paired-end protocol. Exome data analysis was performed using genome analysis tool kit (GATK; www.broadinstitute.org/gatk). For WABS07 and WABS08, whole exome sequencing was performed on DNA, isolated from amniotic fluid cells. Genomic DNA was extracted according to standard procedures and enriched with Agilent Sureselect Clinical Research Exome (CRE) Capture. Samples were run on the HiSeq 4000 (101 bp paired-end, Illumina). On average, 50 million reads per exome and a mapped fraction above 98% were obtained. Average coverage is approximately 50 fold. Reads were mapped to the genome using BWA (bio-bwa.sourceforge.net). Variant detection was performed by GATK. Analysis of variants (VCF file) was performed in Alissa Interpret (Illumina) by filtering on quality (read depth ≥10), minor allele frequency (≥1% in 200 alleles in dbSNP, ESP6500, the 1000 Genome project or the ExAC database), location (within an exon or first/last 10 bp of introns). Variants were further selected based on three inheritance models (de novo autosomal dominant, autosomal recessive and X-linked recessive). Of note, a hemizygous variant of unknown significance (VUS) in the X-linked Duchenne gene DMD, most likely not pathogenic, was found in WABS07 and WABS08. Validation of the identified DDX11 mutations was performed by Sanger sequencing.

**3′RACE PCR**. The second generation 5′/3′ RACE Kit (Roche) was used to amplify the 3′UTR of DDX11 and DDX12p from RPE1 cDNA, by using a non-specific forward primer and a mix of three tagged oligo dT primers. The resulting PCR product was used as a template for a second PCR reaction, using a forward primer specific for DDX11 or DDX12p and a reverse primer that anneals to the tag. The resulting PCR fragments were cloned into zero-blunt plasmids and sequenced using specific primers for DDX11 or DDX12p. Primer sequences are listed in Supplementary Table 1.

**Construction of cell lines and cell culture**. RPE1-hTERT cells (American tissue culture collection) and SV40 transformed fibroblasts, including WABS01[1], WABS02, WABS03, WABS04, WABS05, WABS08, FANCJ patients 1 and 2 (VU1414F and VU030F[78]) and LN9SV control[79], were cultured in Dulbecco's Modified Eagles Medium (DMEM, Gibco), supplemented with 10% FCS, 1 mM sodium pyruvate and antibiotics. Epstein-Barr virus (EBV) transformed lymphoblasts, including WABS01-L, WABS03-L, WABS04-L, WABS05-L, RBS-L (ESCO2 deficient), VU867-L (deficient in both FANC-A and FANC-M)[80], and HSC93 control[81], were cultured in Roswell Park Memorial Institute (RPMI, Hyclone) supplemented with 10% FCS, 1 mM sodium pyruvate and antibiotics.

WABS03-L, WABS04-L, and WABS05-L were functionally complemented by transfection of pMEP4 plasmid (Invitrogen) expressing wild type DDX11 (transcript variant 1, NM_030653.4) and selected with 70 µg/ml hygromycin-B. WABS02 and WABS08 fibroblasts were stably transfected with pIRES-Neo plasmids containing wtDDX11 or (in the case of WABS01) different DDX11 mutants, that were constructed using overlap extension PCR, and selected with 700 µg/mL G418. For DDX11-K50R studies, RPE1-TP53KO-DDX11KO and WABS01 cells were transduced with Lenti-CMVie-IRES-Blast (gift from Ghassan Mouneimne (Addgene plasmid #119863)[82] and selected with 15 µg/mL Blasticidin.

CRISPR-Cas9 was used to construct clonal and transient knockouts in RPE1 cells. The generation of RPE1-hTERT_TetOn-Cas9_TP53KO is described before[53]. Briefly, Cas9 cDNA was cloned into the pLVX-Tre3G plasmid (Clontech) and lentiviral Tre3G-Cas9 and Tet3G particles were produced in HEK293T cells using the Lenti-X HT packaging system (Clontech). Transduced cells were selected with 10 µg/mL puromycin and 400 µg/mL G418. Cells were treated with 100 ng/mL doxycycline (Sigma-Aldrich) to induce Cas9 expression and transfected with 10 nM synthetic crRNA and tracrRNA (Dharmacon or IDT) using RNAiMAX (Invitrogen). The following crRNA sequences were used: TP53 CCATTGTTC AATATCGTCCG, DDX11/12p CTTTGGCAAGGATGTTCGGC,DDX11 specific GGCTGGTCTCCCTTGGCTCC, DDX12p specific GCTTATCAACGACC GCTGCG, OR10A7 AGAAGAGGGACCACCAAATG, POLR2L ACAAGTGGGA GGCTTACCTG, FANCJ#1 AGATTACTAGAGAGCTCCGG, FANCJ#2 TAACA TCGAGGACTGTGCTC, BLM TACGGCCACAGCTAATCCCA, WRN, TAGG AATTGAAGGAGATCAG. Indels were assessed by Sanger sequencing. To specifically amplify DDX11 and DDX12p genomic regions surrounding gRNA target sites, primers inside or spanning a DDX11 specific intronic region were designed (Fig. 5c). Primer sequences are listed in Supplementary Table 1.

**Proliferation assays**. Lymphoblasts were seeded in T25 flasks with increasing drug concentrations, and counted again when untreated cells reached approximately three population doublings. IC50 values were determined using curve fitting (www.ic50.tk). For adherent cells, the IncuCyte Zoom instrument (Essen Bioscience) was used. RPE1 cells (1500/well) and fibroblasts (3000/well) were seeded in 96-wells plates and imaged every 4 h with a ×10 objective. IncuCyte software was used to quantify confluence from four non-overlapping bright field images per well, for at least three replicate wells. Doubling time was calculated for the period required to grow from approximately 30 to 70% confluence, using the formula doubling time (h) = required time (h) * log(2)/(log(confluence endpoint (%))− log(confluence starting point(%))). For clonogenic survival assays, cells were counted and seeded at 1500 cells/well in 6-wells plates. Ten days after seeding, cells were fixed in methanol, and stained in 0.5% crystal violet and 20% methanol.

**siRNA experiments**. For knockdown experiments, 25 nM siRNA (Dharmacon) was transfected using RNAiMAX (Invitrogen). Sequences: non-targeting siRNA UAAGGCUAUGAAGAGAUAC; siDDX11 GCAGAGCUGUACCGGGUUU, CGGCAGAACCUUUGUGUAA, GAGGAAGAACACAUAACUA, UGUUCAA GGUGCAGCGAUA; sip53 GAAAUUUGCGUGUGGAGUA, GUGCAGCUGUG GGUUGAUU, GCAGUCAGAUCCUAGCGUC, GGAGAAUAUUUCACCCU

UC; siUBB CCCAGUGACACCAUCGAAA, GACCAUCACUCUGGAGGUG, GUAUGCAGAUCUUCGUGAA, GCCGUACUCUUUCUGACUA

**qRT-PCR**. Total RNA was extracted with the High Pure Isolation Kit (Roche) and cDNA was prepared with the iScript cDNA Synthesis Kit (Biorad). Quantitative reverse transcription polymerase chain reaction (qRT-PCR) was performed using SYBR Green (Roche) on a LightCycler 480 (Roche). Levels were normalized to the geometric mean of at least two housekeeping genes. Primer sequences are listed in Supplementary Table 1.

**Immunoblotting**. Cells were lysed in lysis buffer (50 mM Tris-HCl pH 7.4, 150 mM NaCl, 1% Triton X-100) with protease- and phosphatase inhibitors (Roche), except for the WABS fibroblasts (Fig. 3a, c), which were directly scraped in sample buffer. Proteins were separated by 3–8%, 4–15% or 8–16% SDS-PAGE (NU-PAGE or BioRad) and transferred to immobilon-P membranes (Millipore). Membranes were blocked in 5% dry milk in TBST-T (10 mM Tris-HCl pH 7.4, 150 mM NaCl, 0.04% Tween-20), incubated with primary and peroxidase-conjugated secondary antibodies (DAKO Glostrup, Denmark; 1:10000) and bands were visualized by chemoluminescence (Amersham). Antibodies used for detection are mouse-anti-DDX11 (B01P, Abnova; 1:250–1:1000), goat-anti-β-actin (I-19, Santa Cruz; 1:1000), mouse-anti-α-tubulin (B-5-1-2, Santa Cruz #sc-23948; 1:2000), mouse-anti-CDC6 (Santa Cruz #sc-9964; 1:500), mouse-anti-p62 (D5L7G, cell signaling; 1:1000), mouse-anti-Flag (M2, Sigma; 1:10000), mouse-anti-p53 (DO-1, Santa Cruz #sc-126; 1:1000), mouse-anti-vinculin (H-10, Santa Cruz #sc-25336; 1:2000), guinea pig anti-ESCO2 (1:500[83]). Uncropped western blots are provided in Supplementary Fig. 13.

**Analysis of cohesion defects and chromosomal breakage**. Cells were incubated with 200 ng/mL Demecolcin (Sigma-Aldrich) for 20 min (cohesion defect analysis) or 30 min (chromosomal breakage analysis). Cells were harvested, resuspended in 0.075 M KCl for 20 min and fixed in methanol/acetic acid (3:1). Cells were washed in fixative three times, dropped onto glass slides and stained with 5% Giemsa (Merck). Cohesion defects or chromosomal breaks were counted in 50 metaphases per condition on two coded slides. Analysis of breaks was described before[84].

**DNA fiber analysis**. Cells were pulse-labeled with 25 μM chlorodeoxyuridine (CldU) for 20 min, followed by 20 min 250 μM iododeoxyuridine (IdU). Approximately 3000 cells were lysed in 7 μL spreading buffer (200 mM Tris-HCl PH 7.4, 50 mM EDTA and 0.5% SDS). Fibers were spread on Superfrost microscope slides, which were tilted ˷15°, air-dried for several minutes and fixed in methanol:acetic acid (3:1). DNA was denatured with 2.5 M HCl for 75 min, blocked in PBS + 1% BSA + 0.1% Tween20 and incubated for 1 h with rat-anti BrdU (1:500, Clone BU1/75, Novus Biologicals) and mouse-anti-BrdU (1:750, Clone B44, Becton Dickinson). Slides were fixed with 4% paraformaldehyde for 10 min, incubated for 1.5 h with goat-anti-mouse Alexa 488 and goat-anti-rat Alexa 594 (both 1:500, Life technologies) and mounted with Vectashield medium. Images of DNA fibers were taken with a Zeiss AxioObserver Z1 inverted microscope using a ×63 objective equipped with a Hamamatsu ORCA AG Black and White CCD camera. Fiber tract lengths were assessed with ImageJ and μm values were converted into kilobases using the conversion factor 1 μm = 2.59 kb[85,86].

**Immunofluorescence**. Cells were grown on cover slips, fixed in 2% paraformaldehyde for 15 min at RT and subsequently in 70% ice cold EtOH for 1 h. Cells were permeabilized in 0,3% Triton X-100 for 5 min, blocked in 3% BSA and 0,3% Triton X-100 for 45 min, incubated with primary antibody for 1.5 h and secondary antibody for 1 h at RT. Cells were mounted using ProLong™ Gold Antifade Mountant with DAPI (Invitrogen) and cells were examined using fluorescence microscopy (Leica). Antibodies used: rabbit anti-DDX11 (ab204788, Abcam), mouse-anti-yH2AX (Ser139, JBW301, Millipore), rabbit anti-53BP1 (Novus)

**Flow cytometry**. Cells were harvested, washed in PBS and fixed in ice cold 70% EtOH. Cells were washed and resuspended in PBS with 1:10 PI/RNase staining buffer (BD Biosciences) and analyzed by flow cytometry on a BD FACSCalibur (BD Biosciences). Cell-cycle analysis was conducted with BD CellQuest software (BD Biosciences).

**Recombinant protein production and DNA helicase assay**. DDX11-3xFlag (wild type and the mutant derivatives R140Q and C705Y) were produced in HEK 293T cells transiently transfected with pcDNA 3.0 plasmid constructs and purified as previously described[40]. The following PAGE-purified oligonucleotides, used for the DNA substrate preparation, were purchased from Sigma: fluorescently-labeled D1 oligonucleotide (5′-[6FAM]CTACTACCCCCACCCTCACAACCTTTTTTTTT TTTTT-3′); D3 (5′- TTTTTTTTTTTTTTTGGTTGTGAGGGTGGGGGTAGTAG-3′) and Cap1 (5′- CTACTACCCCCACCCTCACAACC-3′). For preparation of the forked DNA substrate, the D1 oligonucleotide was annealed to a 3-fold molar excess of the D3 oligonucleotide by incubation at 95 ℃ for 5 min and gradually cooled below 30 ℃. DNA helicase assays were carried out in reaction mixtures

(20 μL) containing the indicated proteins in buffer 25 mM Hepes-NaOH pH 7.2, 1 mM MgCl$_2$, 25 mM K-acetate, 1 mM dithiothreitol, 0.1 mg/mL BSA, 1 mM ATP, 10 nM DNA substrate (D1:D3 annealed oligonucleotides) and 100 nM Cap1. Assays were initiated by addition of the indicated proteins and then incubated for 20 min at 37 ℃. Reactions were quenched with the addition of 5 μl of 5 x stop solution (0.5% [w:v] SDS, 40 mM EDTA, 0.5 mg/ ml proteinase K, 20% [v:v] glycerol). Samples were run on a 8% polyacrylamide-bis (29:1) gel in TBE containing 0.1% (w:v) SDS at a constant voltage of 100 V. After the electrophoresis, gels were analyzed using an imaging system (VersaDoc, BioRad Laboratories) instrument. Displaced oligonucleotide was quantified using the ImageJ program (version 1.52) and any free oligonucleotide in the absence of protein was subtracted. The enzymatic assays were carried out in triplicate.

**Generation of DDX11-G57R mice**. A 25 residue single-stranded DNA oligonucleotide was designed to substitute a single nucleotide in the endogenous *DDX11* gene in mouse embryonic stem (ES) cells to exchange glycine codon 57 for an arginine codon. Oligonucleotide-directed gene modification was performed essentially as described[87]. Mutant mice were generated by injecting mutant ES cells into C57Bl/6 blastocysts following standard procedures. All animal study protocols were approved by the NKI Animal Welfare Body. Mutant and wild-type DDX11 alleles were distinguished using mutant- and wild-type-specific PCR primers. Primer sequences are listed in Supplementary Table 1.

**Reporting summary**. Further information on research design is available in the Nature Research Reporting Summary linked to this article.

## Data availability
DDX11 sequencing reads from WABS patients have been deposited to the Sequence Reads Archive (SRA) with accession code PRJNA645773. For sequence alignments, we used Homo sapiens chromosome 12, GRCh38.p13 Primary Assembly (accession NC_000012.12, which contains both DDX11 and DDX12p gene sequences; as well as DDX11 transcript, accession NM_030653.4, and DDX12p transcript, accession NR_033399.1. The here obtained sequence of the DDX12p transcript of RPE1-TERT cells has been deposited to GenBank, and is available with accession code MT747418. All other relevant data supporting the key findings of this study are available within the article and its Supplementary Information files, the source data, or from the corresponding authors upon reasonable request. Source data are provided with this paper.

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

## Acknowledgements
The authors thank Henri van de Vrugt for the RPE1-hTERT_TetOn-Cas9 cells and for many helpful comments on the manuscript, Charlotte Pennings, Mark Grannetia and Pascal Walther for helping to construct and validate DDX knockout cells, Govind Pai for his help with DNA fiber assays, Khashayar Roohollahi for helping with processing NGS sequencing data, John Reynolds and Laura McFarlene-Majeed for analysis of WABS02 cells, Corrie Weemaes, Graciela Lagomarsino and Mariangel Ospitaleche for counseling, clinical data and patient material, Quinten Waisfisz, Amandine Batte, Mireille Tittel-Elmer and Haico van Attikum for insightful discussions and sharing data prior to publication. We thank all our colleagues for valuable feedback and support. This work was supported by the Cancer Center Amsterdam (grant CCA2015-5-25), the Dutch Cancer Society (KWF grant 10701/2016-2, a young investigator grant to JdL), the Netherlands Organisation for Scientific research (NWO TOP-GO grant 854.10.013) and a CR-UK Programme grant (C17183/A23303 to GS). FMP received financial support from a European Union's Horizon 2020 research and innovation program grant Agreement No 859853 (MSCA-ETN 2019, "AntiHelix"), from Consorzio CNCCS ("Progetto B - Ricerca di nuovi farmaci per malattie rare trascurate e della povertà") and from Regione Campania (POR-FESR 2014-2010, "Progetto SATIN"). We dedicate this work to the memory of Johan de Winter, who first identified WABS in our lab.

## Author contributions
J.v.S., A.F., J.B., G.S., A.O., M.R., N.A., M.M., E.C., and J.d.L. carried out the experiments, which were supervised by F.P., H.t.R., R.W., and J.d.L. Patient samples and clinical data were provided by J.P., G.S., C.d.A.E., K.D., M.v.S., I.B., and K.D. The manuscript was written by J.v.S., A.F., R.W., and J.d.L.

## Competing interests
The authors declare no competing interests.
