## [Peer Review File · Nature Communications]

Reviewers' comments:

Reviewer #1 (Remarks to the Author):

DDX11/Chl1 and its yeast orthologue Chl1 are ATP-dependent DNA helicases related to Xpd and FANCD1, that unwinds DNA with 5'-3' directionality. Mutations in DDX11 are linked with Warsaw breakage syndrome (WABS) which is characterised by high incidence of chromosomes with centromeric cohesion defects ("railroads") and "premature chromatid separation" (PCS). It has previously been documented that yeast Chl1 and mammalian DDX11 are associated with the replisome and promote establishment of sister chromatid cohesion during S phase. How they are recruited to the replisome is not fully understood. DDX11 associates with Timeless via a highly conserved amino acid motif embedded in a large CHL1/DDX11-specific loop inserted into the helicase domain (Cortone et al PLOS gen 2018). Mutations in DDX11 that abolish this interaction show cohesion defects similar to DDX11 depletion. It was recently shown that DDX11 and Timeless collaborate in order to detect and process G4 structures during replication (Lerner, LK et al bioRxiv). How DDX11 contributes to cohesion establishment and DNA replication is currently unclear.

There is some conflicting information about the requirement of the helicase activity of DDX11/Chl1 in promoting cohesion establishment. Studies of the yeast and Mammalian Chl1 proteins claim that the Walker A mutant (K48 in yeast and K50 in vertebrates) as well as the Q loop mutant Q23, both of which abolish helicase activity are fully capable of promoting cohesion establishment (Samora et al 2016 and Cortone et al 2018). This has led to the notion that the helicase activity is not required and that a hitherto mysterious activity that requires 'engagement' with cohesin and/or Scc2 is instead the mechanism by which Chl1 promotes cohesion establishment. However, analysis of the same Walker mutant in yeast Chl1 or chicken DDX11 by different groups showed the opposite, namely that the helicase activity was critical for cohesion (Holloway S 2000 and Abe T et al 2016). A recent report also showed that the DDX11 Q23A mutant is incapable of rescuing the cohesion defects in WABS fibroblasts, suggesting that the helicase activity is required. Strangely, this paper is from the same lab represented in this manuscript (Faramarz A et al PLOS one 2020). This issue is an important one, as a requirement for the helicase activity would imply that the helicase promotes cohesion establishment by altering the conformation of DNA. The role of Chl1/DDX11's helicase activity therefore is of substantial interest for the cohesin field.

In the current study, the authors present clinical features of seven new WABS patients and characterise the new cases initially by identifying bi-allelic mutations in these patients. They show cells derived from patients display significant cohesion defects and importantly these defects can be corrected by expression of wild type DDX11. A few of the mutations identified are hypomorphic, presumably because of their low protein stability (which they measure). Two of the mutations (G57R and C705Y) lie in the helicase domain of DDX11, the C705Y mutant is shown to abolish DDX11 helicase activity. The paper describes the generation of DDX11 deletion cell lines (with and without p53) and shows that this is accompanied by severe cohesion defects. They show that overexpression of wild type and the hypomorphic mutants rescue the cohesion defects of the DDX11 knockout lines. In contrast, overexpression of the G57R and C705Y mutants as well as the K50A Walker A mutant fail to rescue the cohesion defects. These observations are striking and support the notion that the helicase activity is necessary for its role in DNA replication/repair and in cohesion establishment. Having said this, would it not have been better to 'knock in' the mutant proteins? If the manuscript had been presented in this manner, it would have warranted publication in Nature communications. However, in its current state we cannot recommend publication of this manuscript.

The main problem the manuscript suffers from is that too many unnecessary details are presented. Most of these (Sensitivity to various drugs, modest reduction in fork speeds, accumulation of damage foci) were already known for WABS cells and the observations presented here do not greatly alter our understanding. Large sections Figures 1-7 could easily be presented as supplements and each discussed with a few lines. The main message of the paper (as prominently

stated in the title) appears in Figures 7 & 8. Any general reader would have to navigate through all the details presented with different cells and drugs and redundant messages to get to the heart of the story causing them to lose interest. Inexplicably, the evidence that the newly identified mutation is defective in helicase activity is presented supplementary Figure S9. We do not see the point of cramming all these details into the manuscript just for the sake of adding 'substance' to the paper, it is very distracting and makes it difficult to read.

Another major issue is the fact that the same group of authors have very recently published a paper with all the same assays and show in that manuscript that the helicase activity of DDX11 (based on the analysis of a Q loop mutant Q23A) is required for cohesion and DNA replication/repair (Faramarz A et al PLOS one 2020). Why make two papers out of it? This is a matter for consideration by the editor.

The manuscript does have an important message. If this manuscript is to be considered further, it needs serious rearrangement and rewriting in emphasizing the main message, namely the requirement of the helicase activity for cohesion establishment. The mutant proteins could be characterised biochemically to show the effect on ATP binding and hydrolysis in order to make a strong case that the helicase activity of DDX11 is important for promoting cohesion establishment.

Reviewer #2 (Remarks to the Author):

Summary:

Researchers identified new compound heterozygous mutations linked to Warsaw Breakage Syndrome. The DDX11 proteins encoded by the variants display reduced protein stability; the corresponding cells are hyper-sensitive to topoisomerase and PARP inhibitors, show defects in sister chromatid cohesion, and reduced replication fork speed. DDX11 depletion causes chromosomal instability. Treatment of DDX11-deficient cells with G4-interactive compounds was reported to impair growth induce chromosomal instability and cohesion defect to a more significant extent than loss of FANCD1, a previously determined FANCD1 G4-resolving helicase sharing sequence homology with DDX11 in its helicase core domain. DDX11 helicase core domain is required for G4 ligand resistance. Based on their experimental results, authors conclude that DDX11 is important for protection against G4-induced DSBs as well as cohesion defects that are linked to DNA replication.

Critical Comments:

As detailed in the Summary, this is an investigation of the molecular defects of new patients afflicted by Warsaw Breakage Syndrome, a DDX11 DNA helicase disorder. The new mutations identified help to solidify the molecular pathology underlying the chromosomal instability disorder. The evidence supports the conclusions made by the authors, and the experimental studies are solid. The results will provide value to researchers and clinicians alike. Statistical analyses are appropriate. Experimental procedures and rationale are explained well.

A number of concerns are raised below that need to be carefully considered and addressed by the authors:

Introduction: Suggest that phrasing "putatively moderate symptoms of human WABS patients" be changed as this is subjective and not well defended.

I am a bit confused over the diagnosis of WABS03 as "undoubtedly a WABS patient" (Discussion). Based on protein expression it is stated that the corresponding mutant protein R140Q encoded by a mutant allele shows a small increase and that evidence it is the disease-causing mutation remain limited (paraphrase, page 7 lines 215-219). Further, immunofluorescence studies suggested that

the R140Q mutant localized to nucleus similar to wt DDX11. Please clarify the basis for assigning WABS03 as a patient with Warsaw Breakage Syndrome so that it is easier for the reader to understand.

Perhaps I am missing it, but were all the DDX11 missense mutant proteins, (e.g., R140Q) tested for helicase activity in vitro? I see in Figure S9 that C 705Y was tested. The authors should test R140Q.

On page 6, the authors suggest that the rationale to assess if WABS cells show a replication defect as assessed by DNA fiber assays was that DDX11 is known to interact with replication factors. While this may be true, a number of these proteins are also implicated in DNA repair pathways. Moreover, a previous paper published in 2016 demonstrated that DDX11-deficient cells display reduced replication fork progression in cells exposed to the nucleotide-depleting agent hydroxyurea compared to DDX11-proficient cells (as demonstrated by DNA fiber studies). I found this reference (39) in the Discussion, but perhaps it should be mentioned earlier, as this reference (39) provides a stronger rationale for investigating if the lymphoblasts of newly identified DDX11 patients display replication defects.

Two G4-stabilizing compounds are used for cellular studies: Quarfloxin and CX5461 (reference 53). I do not find in reference 53 any mention of Quarfloxin. Perhaps I am missing it. That paper does list CX3543 in addition to CX5461. Further reading in the literature revealed to me that Quarfloxin is indeed CX3543. Perhaps this could be better spelled out for the reader.

The greater Quarfloxin sensitivity of WABS cells compared to FANCI-deficient cells is interesting but seems under-explored in this study. Although one could argue that the focus of the current work is on DDX11, I found the analysis by comparison to be perfunctory. For example, why not compare the sensitivity of FANCI- versus DDX11-deficient cells to telomestatin as well? The suggestion that the difference in Quarfloxin sensitivity is based on the observation that "different G4 interacting compounds appear to selectively bind different types of G-quadruplexes" (demonstrated in vitro) may or may not be relevant in vivo and at first glance seems to be speculative. From an experimental standpoint, measuring drug sensitivity by colony survival and acute DNA damage induction by an appropriate immunofluorescent marker (e.g., gamma-H2AX; 53BP1) may provide greater insight than the IncuCyte assay of cell growth for differential effects of the G4 stabilizer as it relates to DDX11 versus FANCI status.

Returning to the point made by the authors that the apparently differential effects of Quarfloxin versus Telomestatin on DDX11-deficient cells versus FANCI-deficient cells may reflect differences in G4 binding specificity, the authors should address this directly in the Discussion by reporting what has been published in the literature on this issue. The mention in the Discussion that TMS stabilizes formation of telomeric G4 structures whereas Quarafloxin selectively targets a G4 structure commonly found in ribosomal DNA and in promoter sequences, and that quarfloxin accumulates in nucleoli. These points raise the question if a particular G4 architecture is favored for the ribosomal DNA repeats housed in the nucleoli versus the telomeric repeats of telomeres? Given that DDX11 was previously reported to efficiently resolve 5' ssDNA tailed two-stranded anti-parallel G4 substrates in vitro but act poorly on unimolecular G4 substrate which FANCI is capable of resolving efficiently, it raises the question following the Reviewer's comments if Quarafloxin preferentially binds bimolecular G4 substrates versus unimolecular G4 substrates, or (as the authors also suggest) that the preferential subcellular localization of Quarafloxin to nucleoli plays a role and suggests a more prominent nucleolar role of DDX11. Is it known what is the predominant G4 architecture of ribosomal DNA repeats?

Given 1) the potential effects of G4-stabilizing compounds on nuclear DNA replication, 2) the observation made in the current study that DDX11-deficient cells show reduced replication speed, and 3) the previously published result that forks are preferentially slowed by hydroxyurea (an agent that like G4 binding cmpds induces replication stress but by a distinct mechanism) in

DDX11-deficient cells compared to Wild-type cells, I would have expected the authors to perform additional DNA fiber analyses to assess effects of the G4-stabilizing agents on not only replication fork speed, but other fork dynamic parameters, i.e., fork asymmetry, nascent strand degradation, fork restart, etc. These additional studies may have yielded insight to the relationships among DDX11 status, G-quadruplexes, chromosomal instability and sister chromatid cohesion, as suggested by the authors. Moreover, the DNA fiber studies with DDX11-deficient and proficient cells exposed to G4-binding compounds need to be done to support the conclusions made in the study and provide greater insight to the apparent G4 ligand sensitivity reported in this manuscript.

In Figure 2 (panels C and D), the authors show a single image of gamma H2AX and DNA fiber assays with double label (ongoing replication fork), respectively. The authors should show representative images for both panels C and D from the various cell lines in which quantitative data are shown.

Similarly, in Figure 3H, representative metaphase spreads for the analyzed cell lines in which quantitative data are shown should be depicted.

Reviewer #3 (Remarks to the Author):

Faramarz and colleagues describe molecular and cellular defects in cells derived from patients with Warsaw Breakage Syndrome (WBS). Previous studies (as well as the current work) link defects in DDX11 to the development of WBS. While mRNA levels are unaffected WBS cells exhibit DDX11 protein instability. Mutations in some patients are also likely to impair the helicase activity of DDX11. In addition to chromosome breakage WBS cells exhibit impaired chromosome cohesion, hyper-sensitivity to DNA damaging agents such as camptothecin and PARP inhibitors, decreased DNA synthesis and elevated chromosomal aberrations. RPE1 cells knocked out for DDX11 also exhibit impaired cell proliferation that is dependent on P53. Importantly they discount the contribution of the pseudogene DDX12p to these phenotypes. Faramarz et al go on to show that DDX11 defective cells are hyper-sensitive to G4-DNA stabilizing compounds which inhibit cell proliferation, induced loss of cohesion and caused increased chromosomal breaks. The same is true for WBS lymphoblasts and the defects were reversed in cells expressing wild type DDX11.

This is a nice piece of work, particularly strong when describing the defects associated with loss of DDX11 and with WBS. However, it is much less strong on mechanism. It documents defects in cohesion and in replication but reports little as to why this is. The biggest limitation is that the manuscript highlights the role of DDX11 in protecting against G-quadrupled induced DNA damage but little about the relevance of this to normal metabolism. A major limitation is the use of G4-stabilizing ligands such as pyridostatin. This compound has recently been shown to trapping of Topoisomerase 2 on DNA but it is not clear whether this is linked to G4 stabilization (<https://www.biorxiv.org/content/10.1101/845446v1>). So it is possible that the DDX11 defective cells are mainly sensitive to trapping of topoisomerases rather than DNA damage induced by persistent G4-DNA structures. Even, so the investigation into the contribution of DDX11 to DNA repair and the DNA damage response is quite superficial, particularly as the title emphasizes this aspect. It would be useful if the authors looked in more detail at the genetic relationship between DDX11 and genes shown to affect response to G4-DNA such as FANCI, REV1, BLM and WRN. Ultimately the reader would like to understand wto which physiological pathway(s) DDX11 contributes.

Reviewer #1 (Remarks to the Author):

DDX11/ChlR1 and its yeast orthologue Chl1 are ATP-dependent DNA helicases related to Xpd and FANCI, that unwinds DNA with 5'-3' directionality. Mutations in DDX11 are linked with Warsaw breakage syndrome (WABS) which is characterised by high incidence of chromosomes with centromeric cohesion defects ("railroads") and "premature chromatid separation" (PCS). It has previously been documented that yeast Chl1 and mammalian DDX11 are associated with the replisome and promote establishment of sister chromatid cohesion during S phase. How they are recruited to the replisome is not fully understood. DDX11 associates with Timeless via a highly conserved amino acid motif embedded in a large CHL1/DDX11-specific loop inserted into the helicase domain (Cortone et al PLOS gen 2018). Mutations in DDX11 that abolish this interaction show cohesion defects similar to DDX11 depletion. It was recently shown that DDX11 and Timeless collaborate in order to detect and process G4 structures during replication (Lerner, LK et al bioRxiv). How DDX11 contributes to cohesion establishment and DNA replication is currently unclear. There is some conflicting information about the requirement of the helicase activity of DDX11/Chl1 in promoting cohesion establishment. Studies of the yeast and Mammalian Chl1 proteins claim that the Walker A mutant (K48 in yeast and K50 in vertebrates) as well as the Q loop mutant Q23, both of which abolish helicase activity are fully capable of promoting cohesion establishment (Samora et al 2016 and Cortone et al 2018). This has led to the notion that the helicase activity is not required and that a hitherto mysterious activity that requires 'engagement' with cohesin and/or Scc2 is instead the mechanism by which Chl1 promotes cohesion establishment. However, analysis of the same Walker mutant in yeast Chl1 or chicken DDX11 by different groups showed the opposite, namely that the helicase activity was critical for cohesion (Holloway S 2000 and Abe T et al 2016). A recent report also showed that the DDX11 Q23A mutant is incapable of rescuing the cohesion defects in WABS fibroblasts, suggesting that the helicase activity is required. Strangely, this paper is from the same lab represented in this manuscript (Faramarz A et al PLOS one 2020). This issue is an important one, as a requirement for the helicase activity would imply that the helicase promotes cohesion establishment by altering the conformation of DNA. The role of Chl1/DDX11's helicase activity therefore is of substantial interest for the cohesin field.

We thank the reviewer for the interest in our work. Indeed, we are similarly intrigued by the question whether or not the helicase activity of DDX11 and topological alterations to the DNA contribute to cohesion establishment. Because several papers have reported conflicting results, we find it important to provide multiple lines of evidence, as presented in our current manuscript.

Importantly, experiments with the DDX11 Q23A mutant described in Faramarz et al (2020)¹ which the reviewer refers to, do not provide sufficient evidence for a specific role of the DDX11 helicase domain in promoting sister chromatid cohesion.

The DDX11 Q23A mutant lacks DNA binding activity² and therefore, a failure to rescue cohesion defects by this mutant, while in agreement with a role for the helicase domain, could still be explained by helicase-independent function of DDX11 that would contribute to cohesion establishment. So, the results with the Q23A mutant, while relevant, provide insufficient proof that the helicase activity of DDX11 *per se* is required for cohesion establishment.

Here we provide the following lines of evidence that the topological changes of DNA and DNA helicase activity of DDX11 indeed contribute to sister chromatid cohesion: 1) treating RPE1 cells with compounds that stabilize G4 structures induces sister chromatid cohesion; 2) DDX11 deficient cells are extremely sensitive to these compounds, and show aggravated cohesion defects; these effects are not observed for other disease associated putative G4 helicases such as FANCI; 3) the K50R mutant DDX11, which resides in the nucleus but specifically affects the helicase domain of DDX11, completely fails to rescue cohesion defects as well as toxicity of the G4 stabilizer; 4) a novel WABS associated C705Y allele of DDX11 identified here, which we demonstrate lacks DNA helicase activity in vitro, also fails to rescue cohesion defects, whereas alleles with reduced protein stability do rescue cohesion defects when overexpressed.

In the current study, the authors present clinical features of seven new WABS patients and characterise the new cases initially by identifying bi-allelic mutations in these patients. They show cells derived from patients display significant cohesion defects and importantly these defects can be corrected by expression of wild type DDX11. A few of the mutations identified are hypomorphic, presumably because of their low protein stability (which they measure). Two of the mutations (G57R and C705Y) lie in the helicase domain of DDX11, the C705Y mutant is shown to abolish DDX11 helicase activity. The paper describes the generation of DDX11 deletion cell lines (with and without p53) and shows that this is accompanied by severe cohesion defects. They show that overexpression of wild type and the hypomorphic mutants rescue the cohesion defects of the DDX11 knockout lines. In contrast, overexpression of the G57R and C705Y mutants as well as the K50A Walker A mutant fail to rescue the cohesion defects. These observations are striking and support the notion that the helicase activity is necessary for its role in DNA replication/repair and in cohesion establishment.

We thank the reviewer for his/her appreciation of our work.

Having said this, would it not have been better to 'knock in' the mutant proteins? If the manuscript had been presented in this manner, it would have warranted publication in Nature communications. However, in its current state we cannot recommend publication of this manuscript.

Although cDNA overexpression (wt versus mutant) is a clinically established and validated approach for the diagnosis of for instance Fanconi anemia and, as we show, Warsaw Breakage Syndrome, we agree that knocking-in mutant DDX11 would be an elegant way to study their effect at endogenous expression levels.

However, in this case we chose to use cDNA expression studies to investigate the function of DDX11 for the following reasons. First, considering the very low DDX11 protein levels in WABS cells, we hypothesized that patient-derived mutations often destabilize DDX11 protein. The study of protein stability is difficult if the basal protein cannot be detected, which was facilitated by the possibility to create overexpression cell lines. Our approach indeed revealed that multiple mutations found in patients reduce DDX11 protein stability, while they do not affect DDX11 activity, which we believe is an important message. Also the investigation of mutant DDX11 protein localization by immunofluorescence required higher levels than could be detected in patient cells. Secondly, CRISPR-based editing of endogenous DDX11 is hampered by the presence of several DDX11-like sequences in the human genome, including (but not limited to) DDX12p, as described in detail in our manuscript (Figure 5, S1, S4, S6 and S7). This would make a knock-in approach extremely challenging, while the added value would be limited.

The main problem the manuscript suffers from is that too many unnecessary details are presented. Most of these (Sensitivity to various drugs, modest reduction in fork speeds, accumulation of damage foci) were already known for WABS cells and the observations presented here do not greatly alter our understanding. Large sections Figures 1-7 could easily be presented as supplements and each discussed with a few lines. The main message of the paper (as prominently stated in the title) appears in Figures 7& 8. Any general reader would have to navigate through all the details presented with different cells and drugs and redundant messages to get to the heart of the story causing them to lose interest. Inexplicably, the evidence that the newly identified mutation is defective in helicase activity is presented supplementary Figure S9. We do not see the point of cramming all these details into the manuscript just for the sake of adding 'substance' to the paper, it is very distracting and makes it difficult to read.

We appreciate the reviewer's concern about the paper's readability and presumed redundancies. We wish to emphasize however that we face slightly conflicting observations around DDX11 in literature, e.g. regarding different DNA damage drug sensitivities and effects on DNA replication, as explained in our Discussion section. We therefore considered it to be of importance to evaluate the effects of the different, independently identified DDX11 alleles in sufficient detail, using the various relevant assays.

As also recognized by reviewer #2 and #3, this manuscript is not only valuable for basic researchers but also for clinicians. Considering the rareness of WABS, the identification and characterization of new WABS patients is not trivial and demands a thorough and complete assessment. With this paper we provide a complete and updated overview of Warsaw Breakage Syndrome characteristics. Moreover, the findings that mutations in WABS patients often reduce DDX11 protein stability while leaving some residual DDX11 activity, that DDX11 deficiency triggers a p53 response, and that the pseudogene DDX12p does not display a redundant role with DDX11, are new and highly relevant for understanding WABS.

Nevertheless, in the revised version, we have now streamlined the text and moved a number of drug sensitivity and DNA damage assays to supplemental figures (now Figure S5).

Another major issue is the fact that the same group of authors have very recently published a paper with all the same assays and show in that manuscript that the helicase activity of DDX11 (based on the analysis of a Q loop mutant Q23A) is required for cohesion and DNA replication/repair (Faramarz A et al PLOS one 2020). Why make two papers out of it? This is a matter for consideration by the editor.

We regret the confusion caused by including the Q23A mutant in our recent PLoS One paper. As mentioned above, DDX11-Q23A is defective in DNA binding², precluding conclusions about the requirement of helicase activity. Here we use the previously characterized K50R mutant³ as well as two novel, patient-derived helicase mutants (G57R and C705Y). The use of such helicase mutants is vital to prove the necessity of DDX11 helicase activity.

The manuscript does have an important message. If this manuscript is to be considered further, it needs serious rearrangement and rewriting in emphasizing the main message, namely the requirement of the helicase activity for cohesion establishment. The mutant proteins could be characterised biochemically to show the effect on ATP binding and hydrolysis in order to make a

strong case that the helicase activity of DDX11 is important for promoting cohesion establishment.

We agree with the reviewer that the necessity of DDX11 helicase activity for cohesion establishment is further supported by *in vitro* helicase assays of additional DDX11 mutants. We now include the assessment of C705Y and R140Q DDX11 alleles (Figure 8F). Following the reviewer's criticism, we made the text more concise and moved some data to supplemental (Figure S5). In addition, to make the manuscript stronger on mechanism, we added new data (Figure 7, S10 and S11) that involve a comparison of DDX11 with other G4 helicases. We feel that we now present our main findings in a more clear and balanced fashion.

Reviewer #2 (Remarks to the Author):

Summary:

Researchers identified new compound heterozygous mutations linked to Warsaw Breakage Syndrome. The DDX11 proteins encoded by the variants display reduced protein stability; the corresponding cells are hyper-sensitive to topoisomerase and PARP inhibitors, show defects in sister chromatid cohesion, and reduced replication fork speed. DDX11 depletion causes chromosomal instability. Treatment of DDX11-deficient cells with G4-interactive compounds was reported to impair growth induce chromosomal instability and cohesion defect to a more significant extent than loss of FANCD1, a previously determined FANCD1 G4-resolving helicase sharing sequence homology with DDX11 in its helicase core domain. DDX11 helicase core domain is required for G4 ligand resistance. Based on their experimental results, authors conclude that DDX11 is important for protection against G4-induced DSBs as well as cohesion defects that are linked to DNA replication.

Critical Comments:

As detailed in the Summary, this is an investigation of the molecular defects of new patients afflicted by Warsaw Breakage Syndrome, a DDX11 DNA helicase disorder. The new mutations identified help to solidify the molecular pathology underlying the chromosomal instability disorder. The evidence supports the conclusions made by the authors, and the experimental studies are solid. The results will provide value to researchers and clinicians alike. Statistical analyses are appropriate. Experimental procedures and rationale are explained well.

We thank the reviewer for his/her appreciation of our work.

A number of concerns are raised below that need to be carefully considered and addressed by the authors:

Introduction: Suggest that phrasing “putatively moderate symptoms of human WABS patients” be changed as this is subjective and not well defended.

We apologize for the confusion and have replaced ‘putatively’ by ‘relatively’ and thank the reviewer for noticing this mistake. We mean here that whereas loss of DDX11 is embryonic lethal in mice, human WABS patients are viable.

I am a bit confused over the diagnosis of WABS03 as “undoubtedly a WABS patient” (Discussion). Based on protein expression it is stated that the corresponding mutant protein R140Q encoded by a mutant allele shows a small increase and that evidence it is the disease-causing mutation remain limited (paraphrase, page 7 lines 215-219). Further, immunofluorescence studies suggested that the R140Q mutant localized to nucleus similar to wt DDX11. Please clarify the basis for assigning WABS03 as a patient with Warsaw Breakage Syndrome so that it is easier for the reader to understand.

The WABS diagnosis of this patient is based on the clinical symptoms, cellular characteristics and, importantly, the observation that the cellular defects are rescued by introduction of wild-type DDX11 cDNA. However, whereas R140Q shows mild protein instability, we could not prove that this is the only pathogenic mutation in the paternal DDX11 allele of this patient. We adapted the text in the manuscript to improve readability.

Perhaps I am missing it, but were all the DDX11 missense mutant proteins, (e.g., R140Q) tested for helicase activity *in vitro*? I see in Figure S9 that C705Y was tested. The authors should test R140Q.

Initially, we only tested the C705Y mutant for its *in vitro* helicase activity. In our revised manuscript, we also include R140Q (now main figure 8F). We show that this mutant is not catalytically dead, in agreement with the observed ability to correct sister chromatid cohesion upon overexpression. Possibly, destabilization of the mutant protein encoded by the affected allele contributes to the disease.

On page 6, the authors suggest that the rationale to assess if WABS cells show a replication defect as assessed by DNA fiber assays was that DDX11 is known to interact with replication factors. While this may be true, a number of these proteins are also implicated in DNA repair pathways. Moreover, a previous paper published in 2016 demonstrated that DDX11-deficient cells display reduced replication fork progression in cells exposed to the nucleotide-depleting agent hydroxyurea compared to DDX11-proficient cells (as demonstrated by DNA fiber studies). I found this reference (39) in the Discussion, but perhaps it should be mentioned earlier, as this reference (39) provides a stronger rationale for investigating if the lymphoblasts of newly identified DDX11 patients display replication defects.

We agree with the reviewer and altered the introductory lines for Figure 2D.

Two G4-stabilizing compounds are used for cellular studies: Quarfloxin and CX5461 (reference 53). I do not find in reference 53 any mention of Quarfloxin. Perhaps I am missing it. That paper does list CX3543 in addition to CX5461. Further reading in the literature revealed to me that Quarfloxin is indeed CX3543. Perhaps this could be better spelled out for the reader.

We agree with the reviewer that it needs to be better clarified that Quarfloxin is the same compound as CX3543, but different from CX5461. We added (CX3543) at the point where we first mention Quarfloxin in the text.

The greater Quarfloxin sensitivity of WABS cells compared to FANCD1-deficient cells is interesting but seems under-explored in this study. Although one could argue that the focus of the current work is on DDX11, I found the analysis by comparison to be perfunctory. For example, why not compare the sensitivity of FANCD1- versus DDX11-deficient cells to telomestatin as well? The suggestion that the difference in Quarfloxin sensitivity is based on the observation that “different G4 interacting compounds appear to selectively bind different types of G-quadruplexes” (demonstrated *in vitro*) may or may not be relevant *in vivo* and at first glance seems to be speculative. From an experimental standpoint, measuring drug sensitivity by colony survival and acute DNA damage induction by an appropriate immunofluorescent marker (e.g., gamma-H2AX; 53BP1) may provide greater insight than the InCuCyte assay of cell growth for differential effects of the G4 stabilizer as it relates to DDX11 versus FANCD1 status.

We thank the reviewer for these remarks and suggestions and now included a more thorough comparison between DDX11 and FANCD1, which is presented in Figure 7, S10 and S11. A reliable source of Telomestatin was unfortunately no longer commercially available and therefore had to be omitted from our comparison. However, we included Quarfloxin, CX5461, Pyridostatin and MMC to perform additional drug tests. Similar as in our initial submission, this revealed the highest MMC

sensitivity in FANCF patient fibroblasts, whereas WABS patient fibroblasts were most sensitive to Quarfloxin and CX5461 (Figure S10). Moreover, we confirmed these observations in RPE1-TP53KO-FANCFKO and RPE1-TP53KO-DDX11KO-FANCFKO cells, in which we also performed clonogenic survival assays and immunofluorescence analyses of DNA damage markers (Figure 7). These experiments also revealed that DDX11KO as well as FANCFKO cells were more sensitive to Pyridostatin, but the effect of DDX11 depletion was most dramatic. We further demonstrate a genetic interaction between DDX11 and FANCF, suggesting these two helicases have compensatory functions in normal proliferation. Finally, we developed an assay for acute gene knockout experiments to compare the effects of multiple disease-associated DNA helicases and analyzed recovery from transient exposure as well as growth during continuous exposure. Again, these experiments demonstrated that DDX11 knockout resulted in enhanced sensitivity to G4 stabilizers Quarfloxin, CX5461 and Pyridostatin as compared to FANCF knockout (Figure S11). These results show that DDX11 has a more prominent role than FANCF in protecting against the investigated G4 stabilizers.

Returning to the point made by the authors that the apparently differential effects of Quarfloxin versus Telomestatin on DDX11-deficient cells versus FANCF-deficient cells may reflect differences in G4 binding specificity, the authors should address this directly in the Discussion by reporting what has been published in the literature on this issue. The mention in the Discussion that TMS stabilizes formation of telomeric G4 structures whereas Quarafloxin selectively targets a G4 structure commonly found in ribosomal DNA and in promoter sequences, and that quarfloxin accumulates in nucleoli. These points raise the question if a particular G4 architecture is favored for the ribosomal DNA repeats housed in the nucleoli versus the telomeric repeats of telomeres? Given that DDX11 was previously reported to efficiently resolve 5' ssDNA tailed two-stranded anti-parallel G4 substrates in vitro but act poorly on unimolecular G4 substrate which FANCF is capable of resolving efficiently, it raises the question following the Reviewer's comments if Quarafloxin preferentially binds bimolecular G4 substrates versus unimolecular G4 substrates, or (as the authors also suggest) that the preferential subcellular localization of Quarafloxin to nucleoli plays a role and suggests a more prominent nucleolar role of DDX11. Is it known what is the predominant G4 architecture of ribosomal DNA repeats?

We thank the reviewer for these interesting thoughts. Indeed, both G4 helicases and G4 binding compounds most likely have specific substrate preferences, which could relate to both type and location of the G4 structure. It has in fact been argued that G4 motifs in ribosomal DNA preferentially fold into intermolecular G-quadruplexes⁴. We have now further extended our discussion on this topic (paragraph 3) and we feel that this has improved the interpretation of our results.

Given 1) the potential effects of G4-stabilizing compounds on nuclear DNA replication, 2) the observation made in the current study that DDX11-deficient cells show reduced replication speed, and 3) the previously published result that forks are preferentially slowed by hydroxyurea (an agent that like G4 binding cmpds induces replication stress but by a distinct mechanism) in DDX11-deficient cells compared to Wild-type cells, I would have expected the authors to perform additional DNA fiber analyses to assess effects of the G4-stabilizing agents on not only replication fork speed, but other fork dynamic parameters, i.e., fork asymmetry, nascent strand degradation, fork restart, etc. These additional studies may have yielded insight to the relationships among DDX11 status, G-quadruplexes, chromosomal instability and sister chromatid cohesion, as suggested by the authors. Moreover, the DNA fiber studies with DDX11-deficient and proficient cells exposed to G4-binding

compounds need to be done to support the conclusions made in the study and provide greater insight to the apparent G4 ligand sensitivity reported in this manuscript.

Following the reviewer's suggestion, we have now investigated both fork speed and fork asymmetry in DDX11-WT and in DDX11KO cells, and also investigated fork dynamics upon treatment with Quarfloxin (Figure S5C, Figure 6F, Figure S9A and Figure S9B). These experiments reveal a small decrease of fork speed and increase of fork asymmetry upon DDX11 depletion as well as upon Quarfloxin treatment, suggesting that replication forks stall more frequently in these conditions.

In Figure 2 (panels C and D), the authors show a single image of gamma H2AX and DNA fiber assays with double label (ongoing replication fork), respectively. The authors should show representative images for both panels C and D from the various cell lines in which quantitative data are shown.

We added examples of the different conditions that were assessed.

Similarly, in Figure 3H, representative metaphase spreads for the analyzed cell lines in which quantitative data are shown should be depicted.

We want to stress that different metaphases from the same dish often display variable degrees of cohesion defects, underscoring the need to include sufficient numbers. For this particular experiment, we evaluated 150 metaphases (50 per experiment, in triplicate, scored in a double blind manner) for each of ten different conditions. This resulted in modest but meaningful differences as presented in the bar graph. In our view, it would not be advisable to cherry-pick 'representative' images from each condition that match the observations as shown in the final quantification. Nevertheless, in case the editor demands such example metaphase spreads, we will be happy to supply multiple images as supplemental information.

Reviewer #3 (Remarks to the Author):

Faramarz and colleagues describe molecular and cellular defects in cells derived from patients with Warsaw Breakage Syndrome (WBS). Previous studies (as well as the current work) link defects in DDX11 to the development of WBS. While mRNA levels are unaffected WBS cells exhibit DDX11 protein instability. Mutations in some patients are also likely to impair the helicase activity of DDX11. In addition to chromosome breakage WBS cells exhibit impaired chromosome cohesion, hyper-sensitivity to DNA damaging agents such as camptothecin and PARP inhibitors, decreased DNA synthesis and elevated chromosomal aberrations. RPE1 cells knocked out for DDX11 also exhibit impaired cell proliferation that is dependent on P53. Importantly they discount the contribution of the pseudogene DDX12p to these phenotypes. Faramarz et al go on to show that DDX11 defective cells are hyper-sensitive to G4-DNA stabilizing compounds which inhibit cell proliferation, induced loss of cohesion and caused increased chromosomal breaks. The same is true for WBS lymphoblasts and the defects were reversed in cells expressing wild type DDX11.

This is a nice piece of work, particularly strong when describing the defects associated with loss of DDX11 and with WBS.

We thank the reviewer for the appreciation of our work.

However, it is much less strong on mechanism. It documents defects in cohesion and in replication but reports little as to why this is. The biggest limitation is that the manuscript highlights the role of DDX11 in protecting against G-quadrupled induced DNA damage but little about the relevance of this to normal metabolism.

We agree that it is still difficult to pinpoint the exact mechanisms by which DDX11 supports sister chromatid cohesion, possibly because DDX11 and its DNA substrates contribute to both DNA repair as well as affect cohesin dynamics, leading to a complex picture which we aim to address in more detail in future studies. Nevertheless, we find it striking that in most assays, G4 stabilization has a largely overlapping effect as observed after deletion of DDX11, which we believe helps the field forward to elucidate the roles of DDX11. Our findings indicate that unresolved G4 structures and DNA topology are indeed an important sources of DNA damage linked to cohesion loss in DDX11 deficient cells, even if DDX11 probably targets additional substrates as well. It would indeed be interesting to quantify the amount of unresolved G4 structures in DDX11 depleted cells and to characterize the exact structures that can be *in vivo* substrates of DDX11, but this requires validation of multiple novel assays and future work. We now discuss these matters in more detail in paragraphs 4 and 5 of the Discussion.

A major limitation is the use of G4-stabilizing ligands such as pyridostatin. This compound has recently been shown to trapping of Topoisomerase 2 on DNA but it is not clear whether this is linked to G4 stabilization (<https://www.biorxiv.org/content/10.1101/845446v1>). So it is possible that the DDX11 defective cells are mainly sensitive to trapping of topoisomerases rather than DNA damage induced by persistent G4-DNA structures.

We agree with the reviewer that a putative topoisomerase trapping effect as suggested by this preliminary study could similarly cause a stronger DDX11 dependent response. It seems to us that Pyridostatin has some nonspecific effects; in several experiments, the difference in Pyridostatin

response between DDX11 proficient and DDX11 deficient cells is rather small (Figure 6D, Figure S10, Figure S11, Figure S12A). However, in a new experiment (Figure 7) we find that not only DDX11KO cells, but also cells depleted of the putative G4 helicase FANCD1 are sensitive to pyridostatin. Importantly, we performed most of our experiments using Quarfloxin and CX5461, two compounds that have little structural similarity to Pyridostatin and show different responses in FANCD1 and DDX11 deficient cells. Therefore, we consider it unlikely that the effects of multiple different compounds, that were independently shown to stabilize G4 structures and show dependencies on different genes, can be attributed to a shared off-target effect.

Even, so the investigation into the contribution of DDX11 to DNA repair and the DNA damage response is quite superficial, particularly as the title emphasizes this aspect. It would be useful if the authors looked in more detail at the genetic relationship between DDX11 and genes shown to affect response to G4-DNA such as FANCD1, REV1, BLM and WRN. Ultimately the reader would like to understand to which physiological pathway(s) DDX11 contributes.

It is indeed very interesting to investigate genetic interactions between different G4 helicases and to compare how they affect G4 stabilizer responses. To address these questions of the reviewer in more depth, we have now performed a number of new experiments, which are presented in Figure 7, Figure S10 and Figure S11. These experiments show that DDX11 depletion results in a greater sensitization to G4 stabilizers Quarfloxin, CX5461 and Pyridostatin as compared to FANCD1 knockout. Targeting the DNA helicases WRN and BLM showed that WRN did not affect drug responses, while we only found a mild sensitization following BLM knockout. A genetic interaction was found between DDX11 and FANCD1, suggesting partially compensatory functions between these related Fe-S helicases. Moreover, targeting BLM in DDX11KO cells appeared to reduce growth speed and to have an additive effect on various drug treatments. We conclude that DDX11 is the dominant DNA helicase required to resolve structures that are targeted by the used G4 stabilizers. Additional insights in the molecular contexts of DDX11 activity, the exact conditions within which this becomes relevant and potential overlap with other helicases will require novel research projects.

References

1. Faramarz, A. *et al.* Non-redundant roles in sister chromatid cohesion of the DNA helicase DDX11 and the SMC3 acetyl transferases ESCO1 and ESCO2. *PLoS One* **15**, e0220348 (2020).
2. Ding, H., Guo, M., Vidhyasagar, V., Talwar, T. & Wu, Y. The Q Motif Is Involved in DNA Binding but Not ATP Binding in ChIR1 Helicase. *PLoS One* **10**, e0140755 (2015).
3. Wu, Y., Sommers, J., Khan, I., J., d.W. & Brosh, R. Biochemical characterization of Warsaw Breakage Syndrome helicase. *J. Biol. Chem* (2011).
4. Wallgren, M. *et al.* G-rich telomeric and ribosomal DNA sequences from the fission yeast genome form stable G-quadruplex DNA structures in vitro and are unwound by the Pfh1 DNA helicase. *Nucleic Acids Res* **44**, 6213-31 (2016).

REVIEWERS' COMMENTS:

Reviewer #1 (Remarks to the Author):

Having read the revised version, we are convinced that the manuscript warrants publication in nature communications. This study addresses a key issue in the cohesin field. There have been conflicting views about the role DDX11 plays in promoting cohesion establishment during replication. The detailed characterisation of the newly identified mutations currently presented in figure 8 makes a strong case for the requirement of the helicase activity of DDX11 in order to promote cohesion establishment. This is an important observation that alters the outlook for future understanding of the molecular mechanism of cohesion establishment during replication.

Our concerns were not intended to belittle the authors efforts in any way, we understand that the study is also of interest to clinicians. Comprehensive analysis of independently identified mutations associated with WABS and clarification of conflicting effects reported in literature is very important. Our view was that the way the manuscript was initially presented was distracting for a general reader and more importantly, not enough emphasis was given to the main message of the manuscript, namely the importance of the helicase activity of DDX11 in promoting the establishment of sister chromatid cohesion. Both our concerns are satisfactorily addressed in the revised manuscript and we fully support the immediate publication of the manuscript in nature communications.

Reviewer #2 (Remarks to the Author):

I have carefully read the responses to the Reviewers' comments, including my own. It is my belief that the authors have addressed the great majority of the comments in a satisfactory manner. Importantly, the revised manuscript is significantly improved.

Minor comment: References 19 and 79 in the reference list are the same. This mistake should be corrected.

Minor comment: In addition, Reference 22 has not spelled out correctly the name of the late Johan de Winter, a leading pioneer in the discovery and characterization of Warsaw Breakage syndrome. This error should be corrected. Reference 22 also lacks volume and page numbers.

Minor comment: Reference 23 lacks volume and page numbers.

Minor comment: Reference 38 lacks volume and page numbers.

Minor Comment: Reference 75 lacks volume and page numbers.

Minor comment: Reference 87 lacks volume and page numbers.

There may be additional errors in the reference list. The reference list needs to be carefully examined and corrections made.

Reviewer #3 (Remarks to the Author):

I have looked at the revisions and the justifications provided, including issues raised by other reviewers. My criticism that there is still lack of mechanistic insight as to how DDX11 functions in cohesion, which the authors do acknowledge. Nevertheless, this is a decent piece of work with something of interest to say and the authors have added additional experiments to support their case. In this respect they have addressed my concerns satisfactorily.

Response to reviewers

We thank all reviewers for their thorough evaluation of our work. Undoubtedly, their questions and suggestions have greatly contributed to the quality of our paper and we are very pleased that our work was found suitable for publication in Nature Communications.

REVIEWERS' COMMENTS:

Reviewer #1 (Remarks to the Author):

Having read the revised version, we are convinced that the manuscript warrants publication in nature communications. This study addresses a key issue in the cohesin field. There have been conflicting views about the role DDX11 plays in promoting cohesion establishment during replication. The detailed characterisation of the newly identified mutations currently presented in figure 8 makes a strong case for the requirement of the helicase activity of DDX11 in order to promote cohesion establishment. This is an important observation that alters the outlook for future understanding of the molecular mechanism of cohesion establishment during replication.

Our concerns were not intended to belittle the authors efforts in any way, we understand that the study is also of interest to clinicians. Comprehensive analysis of independently identified mutations associated with WABS and clarification of conflicting effects reported in literature is very important. Our view was that the way the manuscript was initially presented was distracting for a general reader and more importantly, not enough emphasis was given to the main message of the manuscript, namely the importance of the helicase activity of DDX11 in promoting the establishment of sister chromatid cohesion. Both our concerns are satisfactorily addressed in the revised manuscript and we fully support the immediate publication of the manuscript in nature communications.

We are very grateful for the reviewer's efforts to help us presenting our key findings in the best possible way. We agree that the suggested alterations significantly improved the balance of the manuscript to interest the general readers of Nature Communications.

Reviewer #2 (Remarks to the Author):

I have carefully read the responses to the Reviewers' comments, including my own. It is my belief that the authors have addressed the great majority of the comments in a satisfactory manner. Importantly, the revised manuscript is significantly improved.

Minor comment: References 19 and 79 in the reference list are the same. This mistake should be corrected.

Minor comment: In addition, Reference 22 has not spelled out correctly the name of the late Johan de Winter, a leading pioneer in the discovery and characterization of Warsaw Breakage syndrome. This error should be corrected. Reference 22 also lacks volume and page numbers.

Minor comment: Reference 23 lacks volume and page numbers.

Minor comment: Reference 38 lacks volume and page numbers.

Minor Comment: Reference 75 lacks volume and page numbers.

Minor comment: Reference 87 lacks volume and page numbers.

There may be additional errors in the reference list. The reference list needs to be carefully examined and corrections made.

We thank the reviewer for taking the time for his/her very careful evaluation of our paper, both scientifically and on such a detailed level. We corrected the errors in the reference list as suggested.

Reviewer #3 (Remarks to the Author):

I have looked at the revisions and the justifications provided, including issues raised by other reviewers. My criticism that there is still lack of mechanistic insight as to how DDX11 functions in cohesion, which the authors do acknowledge. Nevertheless, this is a decent piece of work with something of interest to say and the authors have added additional experiments to support their case. In this respect they have addressed my concerns satisfactorily.

We thank the reviewer for all the valuable comments and suggestions to gain more insight into the mechanism of DDX11 role in cohesion. Indeed, additional questions still remain, but we are confident that we and others will be able to further build upon this work in future research.